# The Atmospheric Sounder Spectrometer by Infrared Spectral Technology (ASSIST): Instrument design and signal processing

Vincent Michaud-Belleau[1], Michel Gaudreau[1], Jean Lacoursière[2], Éric Boisvert[1], Lalaina Ravelomanantsoa[1], David D. Turner[3], and Luc Rochette[1]

[1]LR Tech, Inc., Lévis, QC, Canada
[2]Consultant, Québec, QC, Canada
[3]NOAA Global Systems Laboratory, Boulder, CO, USA

**Correspondence:** Luc Rochette (luc.rochette@lrtech.ca)

**Abstract.** The Atmospheric Sounder Spectrometer by Infrared Spectral Technology (ASSIST) is a Fourier-transform spectrometer designed, fabricated, and sold by LR Tech Inc., which operates in the thermal infrared. When attached to its automated radiometric calibration module, it functions as an infrared spectroradiometer (IRS) that passively measures the absolute spectral radiance within a 46 mrad full field of view and over the 525 to 3300 $cm^{-1}$ (3 to 19 μm) spectral range. For atmospheric studies, the ASSIST IRS is integrated into a mobile enclosure enabling autonomous and reliable operation across a range of environmental conditions. It is typically configured for downwelling radiance measurements (zenith view) at 0.5 $cm^{-1}$ bin spacing, 0.6 $cm^{-1}$ resolution, and 4 $min^{-1}$ sampling rate, closely replicating the behavior of the Atmospheric Emitted Radiance Interferometer (AERI, in rapid-sampling mode), a similar but older IRS. Atmospheric variables affecting the shape of the downwelling thermal infrared radiance spectrum at ground level can be retrieved from the ASSIST high-resolution measurements using dedicated inversion algorithms. This includes the properties of some aerosols and simple clouds, the mixing ratios of trace gases, and the vertical distribution of temperature and water vapor (thermodynamic profile) in the lower troposphere above the instrument. Due to the form of the radiative transfer equation, thermodynamic profiles can only be retrieved with low to moderate vertical resolution but with sufficient accuracy and temporal resolution to help fill the current boundary layer observational gap. This paper provides a detailed description of the ASSIST design and near real-time processing algorithm producing the calibrated radiance spectra that are useful in a variety of applications.

## 1 Introduction

Observations of spectrally resolved downwelling radiance in the thermal infrared, that is, between 3 and 19 μm, have been performed by dedicated ground-based instruments for at least the last 30 years. Perhaps the most well-known instrument of this type is the Atmospheric Emitted Radiance Interferometer (AERI; Knuteson et al. (2004a, b)), an infrared spectroradiometer (IRS) built around a commercial Fourier-transform spectrometer (FTS) that has been deployed to locations around the world, but other similar instruments have also been used (e.g. Shaw et al. (1995); Rathke et al. (2002); Choi and Seo (2024)). These observations were initially made to evaluate and improve spectral radiative transfer models (Han et al., 1997; Turner et al., 2004; Mlawer and Turner, 2016). Subsequent improvements to line-by-line radiative transfer models, based on both ground-

based infrared radiance measurements made by the Department of Energy's Atmospheric Radiation Measurement (ARM) program (Mather and Voyles, 2013) and complementary observations made by spaceborne infrared spectrometers, opened up a range of other applications as accurate radiative transfer models are critical in most remote sensing applications (Maahn et al., 2020).

One of the strengths of the thermal infrared is that it contains the absorption bands of important atmospheric gases in addition to the spectral signatures of liquid water, ice, and aerosols. This enables retrievals of several tropospheric parameters from the downwelling spectral radiance passively measured by a ground-based IRS: thermodynamic profile (temperature and water vapor) (Feltz et al., 1998; Smith et al., 1999; Turner and Löhnert, 2014), mixing ratios of trace gases such as ozone, carbon monoxide, methane, and nitrous oxide (Yurganov et al., 2010; Mariani et al., 2013; Shams et al., 2022), cloud phase (Turner et al., 2003), optical depth and effective radius for both water and ice clouds (Mace et al., 1998; Turner, 2005; Turner and Eloranta, 2008), and radiative composition and size of aerosols (Turner, 2008; Seo et al., 2022). The ability of a ground-based IRS to provide long-term, well-calibrated radiance observations independent from the diurnal cycle also enables trend detection (Gero and Turner, 2011) and the characterization of the radiative forcing of carbon dioxide (Feldman et al., 2015) and methane (Feldman et al., 2018).

This paper introduces a new ground-based IRS entirely built by LR Tech, the Atmospheric Sounder by Infrared Spectral Technology (ASSIST), which follows design, measurement, and processing philosophies that are similar to those of the AERI (differences between the two instruments will be described forthwith). Like the marine AERI (or M-AERI; Minnett et al. (2001)), the ASSIST has been adapted into a marine version for complementary sea surface temperature and emissivity measurements on a sea-going vessel, but the present article focuses on the standard version of the instrument configured for automated downwelling radiance measurement (zenith view) from the ground. In recent years, this standard version of the ASSIST has been deployed to generate thermodynamic profiles of the atmospheric boundary layer (using the TROPoe retrieval algorithm (Turner and Blumberg, 2018; Adler et al., 2024)) for the SPLASH campaign (de Boer et al., 2023; Adler et al., 2023), the PERiLS project (Kosiba et al., 2024), and other field deployments and experiments (Wagner et al., 2022; Bianco et al., 2024).

The following presentation provides a detailed explanation of the ASSIST's instrument design (Sect. 2) and signal processing algorithm (Sect. 3) through a perspective that is complementary to that of the AERI's instrumentation papers, explaining how the measured interferograms are processed to yield absolute downwelling radiance spectra from 525 to 3300 cm$^{-1}$. For basic validation purposes (Sect. 4), samples of ASSIST spectral radiance are also compared to those measured by a co-located AERI unit and those produced by a radiative transfer model. At the time of this writing, an inversion algorithm is not integrated to the ASSIST's standard signal processing routine and retrievals of tropospheric variables therefore require post-processing efforts whose description falls outside the scope of this presentation. The data shown here were acquired primarily by ASSIST-22 (software v1.10.0), a demonstration unit deployed in Stony Plain AB (53.55° N, 114.11° W) in August to November 2023 and Lévis QC (46.77° N, 71.20° W) in 2024; they are representative of instruments with LR Tech serial numbers ending in V, W, X, and Y (a dozen instruments for production years 2021 through 2024).

## 2  Instrument design

This first section focuses on the ASSIST hardware that autonomously produces the raw science signals and the health monitoring signals in a field-deployable format: the scanning interferometer, the optics, the infrared detector, the conditioning electronics, the calibration module, the environmental enclosure, and the health monitoring system. While the measurement parameters such as spectral resolution and scan velocity are adjustable in the ASSIST, only its default configuration (low scan speed, high spectral resolution), which is most relevant for atmospheric studies, is presented here.

Formally, the standalone spectrometer that can be used for laboratory applications constitutes the ASSIST II, but for simplicity the whole instrument, that is, the ASSIST II spectrometer with standard infrared detector assembly and radiometric calibration module, all housed in a field-deployable and mobile environmental enclosure, is referred to as "the ASSIST" in this paper.

### 2.1  Interferometer

At the core of the ASSIST is a scanning two-beam interferometer of the Michelson type (Fig. 1). Broadband infrared flux from the scene is incident on the upper half of a circular potassium bromide (KBr) beamsplitter (BS) that is thin-film coated on its first (fore) surface to reflect between 40 and 60% of both polarizations in the 500 to 5000 cm$^{-1}$ range. The balance of the incoming radiation is mostly transmitted through the beamsplitter thanks to weak material absorption and weak Fresnel reflection at the glass-air interface (Li, 1976)). Each copy of the incident flux is reflected and deflected by a bare-gold coated hollow cube corner (CC) retroreflector, with apex aligned with the BS center, back to the lower half of the beamsplitter that is thin-film coated on its second (aft) surface. The two copies of the incident infrared radiation are recombined there to interfere.

This interferometer configuration, with the incident infrared beam laterally offset from BS center and CC apex, constitutes a so-called four-port configuration with two distinct input ports and two distinct output ports (degenerate ports separated by the shearing property of the retroreflector) (Peck, 1948). In the ASSIST, a blackened plate at ambient temperature emits thermal infrared radiation in the secondary input port, here called the reference port. The balanced output port (under the reference port) recombines the copies of the incident flux that have both undergone reflection away from and transmission through the beamsplitter and is used for downstream detection. The unbalanced output port (under the primary input port) is unused and terminated.

In such a two-beam interferometer, the condition of interference between the two recombined copies of the incident flux is determined by their optical path difference (OPD) measured as the difference in optical lengths from beamsplitting plane to CC apices and back. Due to the identical thin-film coatings applied to the upper and lower beamsplitter half-moons located on opposite sides of the nominally flat KBr plate (Fig. 2), the OPD is insensitive to dispersion in KBr and thin-film coatings. This self-compensating BS design produces an undispersed optical power signal versus OPD, a signal conventionally called the interferogram (IGM) which is related to the autocorrelation of the incident radiation (and its Fourier pair, the power spectral density, as stated by the Wiener-Khinchin theorem (Goodman, 2015)), presuming that propagation in the arms of

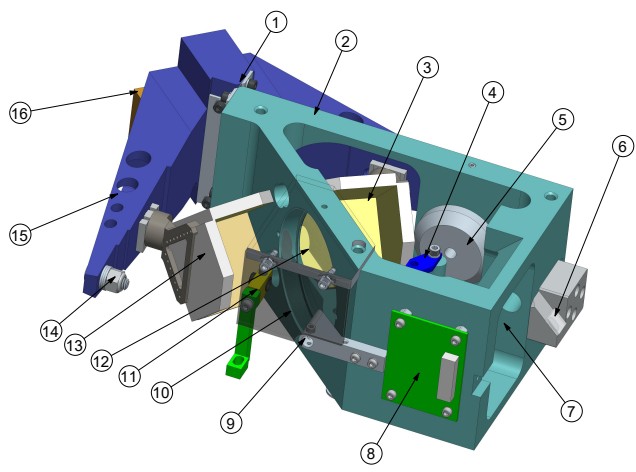

**Figure 1.** Drawing of the scanning interferometer. 1: flexure bearing (double flexure blade) with pivot axis aligned with the beamsplitter median plane. 2: aluminum support structure. 3: fore cube corner retroreflector. 4: white light sensor. 5: voice coil motor. 6: metrology beam injection periscope. 7: primary input port (radiation incident on upper part of the beamsplitter). 8: metrology preamplifier board. 9: white light incandescent lamp. 10: reference port with blackened plate (translucent in drawing). 11: metrology quarter-wave plate. 12: self-compensating KBr beamsplitter. 13: aft cube corner retroreflector. 14: voice coil counterweight. 15: rigid double pendulum structure (wishbone swing arm). 16: pendulum counterweight. In the ASSIST, the interferometer is mounted upside-down relative to this view.

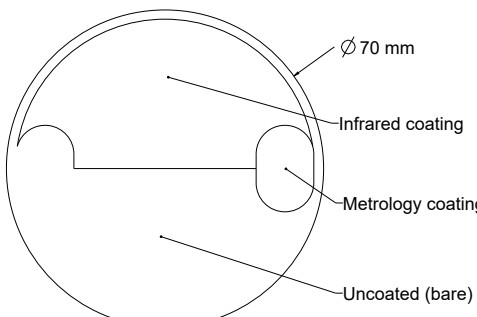

**Figure 2.** Drawing of the KBr beamsplitter as seen from the primary input port. The coating structure is mirrored vertically on the back face.

the interferometer only introduces a differential delay between the split copies of the incident radiation, as the ASSIST is designed to achieve.

    To vary the OPD and trace the interferogram signal, the two cube corners are mounted on a rigid double pendulum structure (Burkert, 1983) that continuously swings in the horizontal plane. This "wishbone" swing arm is balanced such that its center of mass coincides with the pivot axis, minimizing vibration sensitivity and out-of-plane torques. Frictionless, repeatable, and
single-axis rotary motion is enabled by actuation with a voice coil motor and a compliant double flexure blade linking the swing arm to the support structure. The use of retroreflectors naturally compensates for wavefront tilt while wavefront shear is

compensated by mechanical design since the CC apices are mounted at equal distance from the pivot axis (Buijs and McKinnon, 2009). They therefore move on a common circle whose center is designed to be aligned with the BS median plane, reflecting exactly on itself through the beamsplitter. The opto-mechanical gain of this interferometer (the ratio between the OPD and the physical displacement of a retroreflector) is 4 since both retroreflectors are displaced when the swing arm rotates, with one retroreflector moving towards the beamsplitter when the other retroreflector moves away from it.

A linearly polarized helium-neon (He-Ne) metrology laser is used to accurately measure the varying OPD. To this end, its collimated beam is injected into the interferometer parallel to the optical axis such that it hits the lateral edge of the beamsplitter. At both lateral edges and on opposite faces of the BS, thin-film coatings optimized for the nearly monochromatic laser emission at 15 798 cm$^{-1}$ (633 nm) are applied to insular regions (Fig. 2). The laser beam is thus efficiently split, reflected, and recombined like the infrared beam with the exception of transmission through a quarter-wave plate in the aft arm. After recombination, the metrology interferogram is almost purely sinusoidal (narrow optical bandwidth, long coherence length) and the associated fringes are measured by a laser detector (not illustrated: one photodiode and one preamplifier circuit for each laser polarization), providing in-phase and quadrature signals that contain information about the OPD and the sign of the OPD rate of change. These metrology signals serve two purposes: the feedback control of the motor to maintain almost constant OPD scan velocity in the target OPD window (between turnaround events where the scan direction is reversed) and the triggering of the infrared detector's analog-to-digital converter (ADC) on an equally spaced OPD grid that is comparable for forward and reverse scan directions.

The ASSIST is configured to record one sample of the infrared interferogram per metrology fringe at a rate of 32 kHz, which corresponds to an OPD scan velocity of 2.0 cm s$^{-1}$. The total number of interferogram samples at full nominal resolution is 32 768 and the associated OPD span is $\pm1.04$ cm (each cube corner is physically displaced over $\pm0.26$ cm). To ensure symmetry of movement and acquisition around the point of zero path difference (ZPD), an incandescent lamp ("white light") is turned on during the interferometer startup sequence; the metrology fringe that is closest to the maximum of the narrow white-light interferogram is tagged as the point of symmetry until the next startup or initialization (the white light is turned off once the startup sequence is complete). Under normal closed-loop operation of the scanning interferometer, the sampling grid is repeatable from one swing arm scan to the next as there are no fringe count errors (Kleinert et al., 2014). Direct coadding of sampled interferograms, of the same scan direction and for stationary input radiation, can therefore be performed to increase the signal-to-noise ratio. Table 1 summarizes the properties of the interferometer.

## 2.2 Optics

Figure 3 depicts the optical layout of the ASSIST for rays that go through a single arm of the interferometer. In the fore optics segment (upstream of the interferometer), the flux from the scene is redirected using a flat scene mirror (SM) through an antireflection-coated (AR-coated) zinc selenide (ZnSe) entrance window (EW) and towards the primary input port. The scene mirror can rotate about the optical axis and is part of the calibration module described in Sect. 2.5. In the aft optics segment (downstream of the interferometer), the modulated flux is relayed to the infrared detector, described in Sect. 2.3, through the dewar window (DW) using two folding flat mirrors (FM), an off-axis parabolic mirror (OAP), and an AR-coated ZnSe

**Table 1.** Interferometer properties.

| Parameter | Nominal value |
| --- | --- |
| Configuration | Four-port Michelson, 45° AOI |
| Beamsplitter | Self-compensating, coated KBr |
| Reference port | Black plate at ambient temperature |
| Cube corners (CC) | 60 mm clear diam., gold coated |
| Scanning | Flex double pendulum, continuous |
| CC apex motion | Mirrored common circle (no shear) |
| Maximum OPD | $\pm 1.04$ cm |
| OPD scan velocity | 2.0 cm s$^{-1}$ |
| Metrology laser | 633 nm He-Ne, linearly polarized |
| | Coher. length $> 25$ cm, 30 000 h MTTF |

meniscus field lens (FL). All components in the path between the entrance window and the field lens are sealed in a purgeable stainless steel housing with replaceable desiccant to protect the hygroscopic KBr beamsplitter and minimize the absorption of the infrared flux by water vapor.

The ASSIST is a pupil-imaging spectrometer: the aperture stop of the optical train is the detector's photosensitive surface, either that of the square (1-mm side) longwave chip or that of the circular (1-mm diameter) midwave chip. The entrance pupil of the spectrometer, where rays from all fields meet, is located near the CC apices and constitutes a 25-fold magnified image of the photosensitive surface. The shape and size of the spectrometer field of view (FOV) is not determined by the size of the photosensitive surface, but rather by the diameter of the field stop (FS) placed in the OAP's focal plane, an opaque ring that limits the clear diameter of the field lens. As detailed in Sect. 3.4, the shape of the FOV is chromatic, but it can be described to first order and in the far field as a cone with a half-angle of 23 mrad. The spectrometer's geometric étendue is approximately 1.0 mm$^2$ sr in the longwave channel (square pupil) and 0.8 mm$^2$ sr in the midwave channel (circular pupil).

## 2.3 Infrared detector

The photodetection of the modulated infrared signal is performed using a two-color detector in which a mercury cadmium telluride (HgCdTe) photoconductor and an indium antimonide (InSb) photodiode are assembled to detect longwave and midwave radiation, respectively. Instead of the "sandwich" configuration of the AERI (Knuteson et al., 2004a), the ASSIST detector is based on a "dichroic" configuration: in a vacuum dewar, a dichroic beamsplitter (DBS) with approximate cutoff at 1900 cm$^{-1}$ is tilted at 45° relative to the optical axis and steers the longwave beam (transmitted) and the midwave beam (reflected) to the HgCdTe and InSb chips mounted perpendicular to one another (Fig. 3). A cold shield installed inside the dewar minimizes the contribution of out-of-field and unmodulated radiation that produces a constant photocurrent exacerbating shot noise and optical nonlinearities. The dewar is kept at a cryogenic temperature by a slip-on cold finger linked, through a flexible work-

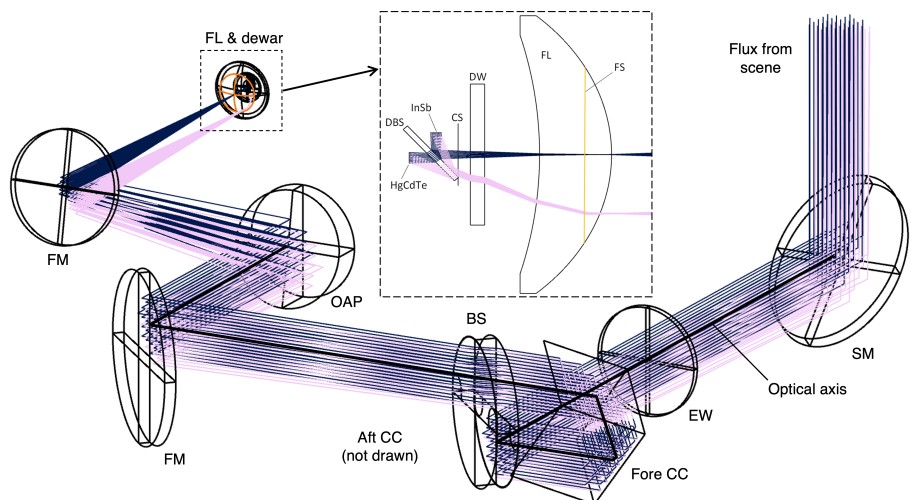

**Figure 3.** Optical layout of the spectrometer. BS: self-compensating KBr beamsplitter. CC: hollow cube corner retroreflectors. CS: cold shield (at dewar temperature). DBS: dichroic beamsplitter or filter. DW: dewar window (AR-coated ZnSe). FL: field lens (AR-coated ZnSe). FM: flat mirror. FS: field stop. HgCdTe: longwave photodetector (1 mm × 1 mm). InSb: midwave photodetector (1 mm diam.). EW: entrance window (AR-coated ZnSe). OAP: off-axis parabolic mirror. SM: scene mirror (rotates about the optical axis). For simplicity, bichromatic rays, with 550 and 1650 cm$^{-1}$ components overlapping until the DBS, are traced for only one arm of the interferometer and from two field points located at infinity: zero and 21 mrad relative to the optical axis (blue and pink, respectively). The air column between calibration plane and photosensitive surface has a length of approximately 124 cm (34 cm open to the atmosphere, 90 cm sealed with desiccant).

ing fluid transfer line that provides some vibration isolation, to a linear split Stirling cooler with flexure bearing compressor design. The cooler is feedback controlled to maintain a default detector temperature of 75.1 K, with fluctuations around the setpoint typically much below 0.1 K in the steady-state regime that is attained approximately 20 min after room-temperature initialization. The detector temperature setpoint can be adjusted to some extent, but operating the detector at this relatively
low temperature enhances the signal-to-noise ratio and sharpens the longwave cutoff in the HgCdTe channel at the expense, presumably, of the nominal 45 000 h mean time to failure (MTTF) quoted at 80.0 K by its manufacturer.

### 2.4  Conditioning electronics

Each raw analog interferogram is processed by a dedicated preamplification and acquisition board. For the HgCdTe channel, this board also provides the stable voltage required to bias the photoconductor and to operate it in the electrically linear "current
mode" (constant positive bias voltage, photocurrent fed to transimpedance amplifier) (Eppeldauer and Novak, 1989). The InSb photodiode is instead zero-biased. The photocurrent produced by the irradiated biased photoconductor or the irradiated photodiode is linearly converted to a voltage (positive in the HgCdTe channel, negative in the InSb channel) by the input stage transimpedance amplifier and then high-pass filtered (AC-coupled) to remove the direct current (DC) component. In the HgCdTe channel, an inverting amplifier increases the alternating current (AC) signal level by a factor of -100. The available

**Table 2.** Spectrometer properties.

| Parameter | Nominal value |
|---|---|
| Optical configuration | Pupil-imaging, no telescope |
| Entrance pupil area | 6.3 / 4.9 cm$^2$ (HgCdTe / InSb) |
| FOV half-angle | 23 mrad (effective, at 1250 cm$^{-1}$) |
| Étendue | 1.0 / 0.8 mm$^2$ sr |
| Detection configuration | Dichroic (cutoff at $\approx$ 1900 cm$^{-1}$) |
| Detector type | Photoconductor / photodiode |
| Sensitivity range (app.) | 500-2050 / 1700-5500 cm$^{-1}$ |
| Detector temperature | 75.1 K |
| Cryogenic cooler | Linear split Stirling (flexure) |
|  | 45 000 h MTTF |
| Interferogram sampling | One sample per metrology fringe |
| Interferogram length | $2^{15}$ samples (forward or reverse) |
| Data rate (incl. turnaround) | 0.95 interferogram s$^{-1}$ |
| Digitization depth | 16 bits (13.5 bits effective) |

bandwidth at that point is typically between 150 and 300 kHz, the exact value depending on the detector unit and detection channel. The AC signal is then further amplified and filtered through a programmable chain to optimize the use of the ADC: the gain is user-adjustable in factor-of-two increments from 1 to 128, and the signal is low-pass filtered by a fourth-order Bessel filter with adjustable cutoff frequency (20 kHz by default) to control signal and noise aliasing. After a final fixed scaling and second high-pass filtering, the conditioned signal is digitized by the 16-bit bipolar ADC triggered at the rising-

edge zero-crossings of the metrology interferogram. The digitized infrared interferogram is finally sent, via serial link, to a timestamping board synchronized to the Global Positioning System (GPS); the board also collects and annotates metadata (settings, scan direction, etc.) and the data produced by the health monitoring system (described in Sect. 2.7). An active GPS antenna with integrated low-noise amplifier is installed on the enclosure roof to receive the GPS signal. Table 2 summarizes the properties of the spectrometer. Figure 4 shows a computer-assisted drawing of the spectrometer assembled in its case

(standalone instrument).

## 2.5  Calibration module

The operation of the ASSIST as a spectroradiometer that measures absolute (or calibrated) downwelling spectral radiance is made possible by the use of a calibration module upstream of the spectrometer. This module is illustrated in Fig. 5 and is built around a flat scene mirror tilted at 45° relative to its rotation axis. Using a stepper motor with a 0.6 mrad microstepping

resolution, this mirror can point the spectrometer FOV towards one of the six view ports of a supporting hexagonal structure. In the standard version of the instrument, only three ports are left open: the two ports at 60° and 300° from zenith are both outfitted

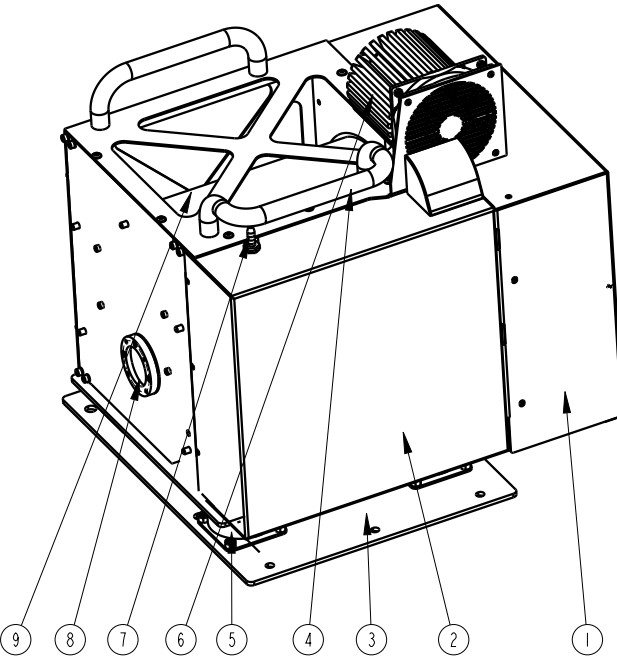

**Figure 4.** Drawing of the spectrometer (formally, the ASSIST II). 1: electronics compartment. 2: aft optics compartment. 3: baseplate (with vibration damping). 4: handles. 5: vibration dampers. 6: heat sink with fan. 7: purge inlet. 8: entrance window. 9: interferometer compartment.

with a reference blackbody (BB) cavity and the zenith port is open to the atmosphere for sky measurements. The scene mirror is coated with unprotected gold to achieve high reflectivity and low polarizance[1] at oblique incidence in the infrared band of interest (Babar and Weaver, 2015). Unprotected gold displays low chemical reactivity, enabling a long lifetime in air, but is extremely soft and easy to damage mechanically. Therefore, to repel particles and hydrometeors from the sensitive scene mirror surface and minimize the need for cleaning interventions, a fan (18 L s$^{-1}$) constantly blows filtered air up at the edge of the zenith port cylinder. Moreover, since an imperfect reflective surface can scatter undesired radiation into the spectrometer FOV, the internal surfaces of the calibration module are all thermally conductive and treated to a matte black finish. An anodized aluminum baffle also rotates with the scene mirror. Stray radiation is therefore approximately scene-independent and removed to first order during radiometric calibration (see Sect. 3.3).

The reference blackbody cavities are close replicas of those used in the AERI (Knuteson et al., 2004a) with a circular aperture open to an internal cone-cylindro-cone geometry (Sapritsky and Prokhorov, 2020). The aluminum internal surfaces are treated with two layers of high-emissivity Aeroglaze Z306 polyurethane coating curing to a matte black finish. Both blackbodies can be precisely temperature-controlled above ambient conditions using integrated resistive heaters, but the normal operation consists in letting one cavity track the ambient temperature in open-loop mode (the ambient blackbody or ABB) and actively

---

[1]The interferometer beamsplitter used at a 45° angle of incidence (AOI) makes the spectrometer's response polarization-dependent. To maintain accurate radiometric calibration, the polarization of the flux entering the interferometer must therefore remain insensitive to the scene mirror orientation.

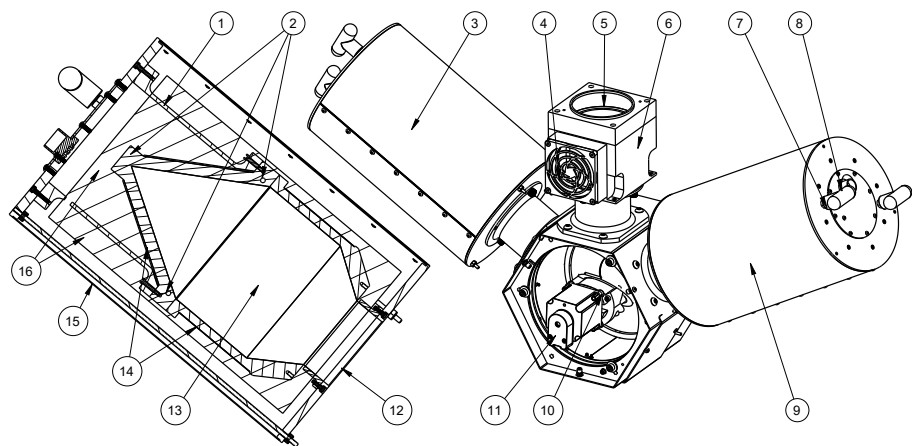

**Figure 5.** Drawing of the calibration module with exposed scene mirror. 1: cavity structural support. 2: cavity thermistors (apex, top, bottom). 3: ambient blackbody (ABB). 4: fan inlet with filter. 5: sky aperture (84 mm diam.) with fan outlet at the edge of the cylinder. 6: removable fan block. 7: blackbody handle. 8: blackbody electrical connector. 9: hot blackbody (HBB). 10: rotating scene mirror. 11: scene mirror stepper motor. 12: cavity aperture (69 mm diam.). 13: coated blackbody cavity. 14: cavity resistive heaters (cylinder and cone for HBB, cylinder only for ABB). 15: blackbody metallic case. 16: blackbody thermal insulation. In operation, a cover seals the motor-side of the hexagonal structure.

maintaining the temperature of the other cavity at 60°C (the hot blackbody or HBB, in closed-loop mode). For this reason, the flexible film heaters are wrapped around both the conical and the cylindrical surfaces of the HBB, but only around the cylinder surface of the typically inactive ABB. To facilitate thermal control, foam sheet insulation is applied between the cavities and their protective stainless steel cases. Near room temperature, the open-loop thermal time constant of the mounted blackbodies is approximately 2 hours.

A dedicated blackbody controller measures the effective temperature of both cavities and feedback-controls the active heaters. The temperatures are monitored using three 46006 super stable glass negative-temperature-coefficient (NTC) thermistors (Measurement Specialties, Inc.) per cavity, all thermally coupled to the emissive surface at key locations within the cavity walls: cone apex, cylinder top, and cylinder bottom (Fig. 5)[2]. To this end, all temperature-dependent resistances are first measured using an integrated dual-range ratiometric ohmmeter: $> 30.9$ kΩ ($< $ -1°C) for the low-temperature range and $< 39.1$ kΩ ($> $ -6°C) for the high-temperature range, with low-temperature-coefficient precision resistors of 500 kΩ and 50 kΩ (0.01 %), respectively, acting as internal references. Each measured resistance $R_j$ ($j = 1, 2, 3, 4, 5, 6$) is then converted by the controller to a temperature $T_j$ using a piecewise Steinhart-Hart (SH) equation (SI units) (Liu et al., 2018):

$$T_j = \left[ A_0 + A_1 \ln R_j + A_3 \left( \ln R_j \right)^3 \right]^{-1}. \tag{1}$$

---

[2]"Top" and "bottom" here refer to the thermistor position along the vertical axis, with the cavity mounted as shown in Fig. 5, and not to position along the cavity symmetry axis.

The $A_0$, $A_1$, and $A_3$ Steinhart-Hart coefficients are read from a lookup table associated to a specific set of thermistor, cable, and controller channel, attempting to capture the behavior of the whole calibration chain (including imperfections such as contact resistance and thermistor self-heating). The six lookup tables needed by a given calibration module are not swappable and are built by LR Tech through a NIST-traceable in-house calibration procedure during which the assembled thermistors are immersed in a circulating glycol bath whose temperature is set to span the -27 to 63°C range with reference to a standard

platinum resistance thermometer. The tables are saved in the blackbody controller's non volatile memory and each contains fitted SH coefficients for 5 K partitions of the -35 to 65°C domain (one set of three SH coefficients valid from 60 to 65°C, one different set of three SH coefficients valid from 55 to 60°C, etc.). The calculated temperatures are finally converted to the Celsius temperature scale, multiplied by 500, and saved as 16-bit signed integers; the discrete temperature grid thus spans ±65.536°C in increments of 2 mK, though the current calibration procedure is only valid above approximately -25°C (Fig. 6).

For control and reporting purposes, the three temperatures associated to a given cavity are weight-averaged by the controller: while the weights are user-adjustable, the default values are 0.8 for the apex thermistor, 0.1 for the top thermistor, and 0.1 for the bottom thermistor. These weights closely replicate the modeled contribution of each cavity surface to the radiance in the cavity aperture (Minnett et al., 2001; Taylor et al., 2020). Section 3.3 provides additional information about the cavity emissivity model and the general radiometric calibration. Onboard temperature regulation is achieved using a software-implemented

proportional-integral-derivative (PID) controller operating at 3.125 Hz. Under closed-loop operation, the controller continuously adjusts the voltage (0 to 24 V) fed to the BB resistive heaters to minimize the temperature error, which is defined as the temperature setpoint minus the weighted mean temperature. This is done independently for the two blackbodies. Though the loop parameters can be adjusted by the user through serial communication with the controller, the PID settings are tuned during instrument production so that a small setpoint step leads to a step response with small overshoot, no oscillation, and a

closed-loop time constant of approximately 2 minutes. Table 3 summarizes the properties of the blackbody cavities.

## 2.6 Environmental enclosure

The spectrometer and its calibration module are housed in a urethane-painted aluminum enclosure enabling autonomous operation under diverse field conditions (Fig. 7). From a thermal standpoint, the spectrometer and associated electronics are shielded from the environment through a combination of thermal insulation (with a double-roof design to minimize direct solar load)

and active control with a modular air conditioning unit that can maintain a steady-state internal temperature close to 32°C for external temperatures up to at least 40°C. Heat generated by the spectrometer and associated electronics is also sufficient to maintain a steady internal temperature for external temperatures above approximately 5°C, although under favorable external conditions this behavior can be observed down to slightly below -5°C (Fig. 8). Otherwise, the internal temperature progressively drops until the trigger temperature is reached, activating the built-in heater which keeps the internal temperature in the 20

to 24°C range. A thermal wall separates the spectrometer cabinet from the calibration cabinet, which itself is not insulated but rather open to the atmosphere via floor vents and the hatch (when open). This allows the ABB's temperature to approximately track the ambient temperature, with unavoidable lag and smoothing caused by thermal filtering (Fig. 9), and enables exposition of the spectrometer's FOV to the atmosphere.

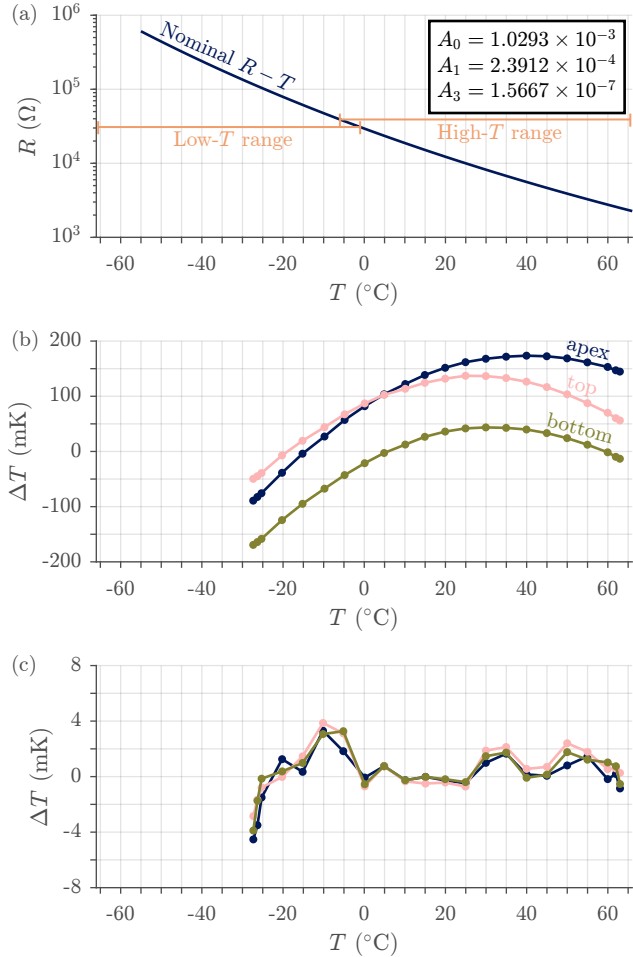

**Figure 6.** Typical thermistor calibration. (a) Nominal thermistor manufacturer $R$-$T$ curve with fitted Steinhart-Hart coefficients given in inset and blackbody controller range limits. Overlapping ranges are used to introduce hysteresis and avoid frequent switching around the transition temperature. (b) Mean temperature error before in-house calibration (controller-measured resistance converted using manufacturer curve minus truth temperature) for ASSIST-22's three HBB thermistors: apex thermistor, top thermistor, and bottom thermistor. (c) Mean temperature error after in-house calibration (calibrated controller readout temperature minus truth temperature) for the same three thermistors. This was measured in a posterior validation test (not a fit residual) and the $\approx 5$ mK peak fitting residuals survive repeated validations with complete disassembly and reassembly of all thermistors.

The motorized hatch is automatically controlled by the precipitation sensor, a heated electrolytic gold-plated surface that is
245 tilted at $45°$ and mounted on a foldable mast that sits next to the hatch. Although automatic operation can be overridden by the user for maintenance or preventive protection, the purpose of the sensor-controlled hatch is to shield the scene mirror and the calibration module from hydrometeors without any user intervention. The ASSIST is programmed to go through the usual measurement cycle when the hatch is closed (instead of stopping to put the scene mirror in a protected position), recording

**Table 3.** Blackbody cavity properties.

| Parameter | Nominal value |
| --- | --- |
| Cavity geometry | Cone-cylindro-cone |
| Aperture diameter | 69 mm |
| Cavity material | Aluminum |
| Internal coating | Aeroglaze Z306 (Lord Corp.), two coats |
| Cavity heating | Resistive film, cone and cylinder for HBB |
| Thermistors | Glass NTC, 10 k$\Omega$ at 25$^\circ$C, 3 per cavity |
| Thermal time constant | 2 h open-loop, 2 min closed-loop |
| ABB temperature | Passively tracks ambient temperature |
| HBB temperature | Actively controlled at 60$^\circ$C |
| $T_j$ measurement range | -65.536 to 65.536$^\circ$C, 2 mK resolution |
| $T_j$ calibration range | -25 to 65$^\circ$C, 5 mK peak fitting residual |

interferograms associated to the infrared emission of the hatch cover's inner surface as seen through the intermediate air path (which can be valuable for monitoring, e.g. Fig. 16). The hatch status, inferred from the readings of two limit switches, is also reported in the output files for filtering purposes. In addition, a webcam is used for independent monitoring of the hatch and its immediate environment; it has proven useful to identify obstacles (spiderwebs, fallen tree branches, packed snow, etc.) in the optical path and also to check the conditions of the sky above the instrument. After a precipitation event, when the sensor surface gets dry enough to send the clear signal (with a programmed 3 min delay), the hatch opens automatically and sky measurements are simply continued.

The enclosure is built on a reinforced bed with forklift-compatible metal rails that facilitate displacement. It can also be moved over short horizontal distances by two people using the puncture-proof (airless) pivoting wheels and the surrounding protective handrail. The total weight of the spectroradiometer in enclosure (with air conditioning unit) is 195 kg and its enclosing envelope L-W-H dimensions, with the mast raised for operation, are 140-102-163 cm. For deployment on uneven surfaces, the enclosure can be lifted and leveled using the integrated stabilizing jacks and spirit level; for field deployment, a grounding electrode accessory can be installed to provide a conductive path between the enclosure chassis and the physical ground for lightning protection. All onboard devices are powered through a single heavy-duty locking power connector (115 or 230 V) and controllable through a single ruggedized locking RJ45 Cat6A Ethernet connector, both rated IP65 and protected from direct precipitation (there is no separate electronics rack; the ASSIST illustrated in Fig. 7 is a standalone instrument). With the air conditioning unit in cooling mode, the ASSIST typically draws 675 W, steady-state. With the air conditioning unit off, the instrument draws at most 300 W.

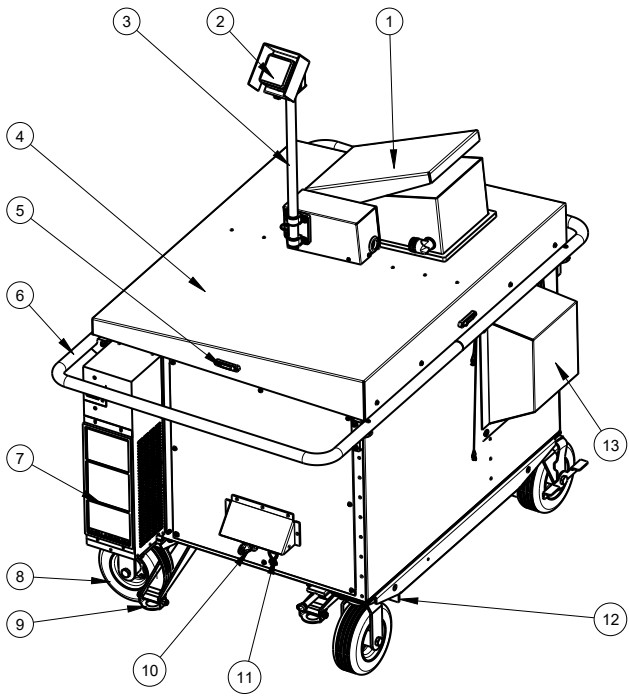

**Figure 7.** Drawing of the environmental enclosure with raised mast and semi-closed hatch. 1: pivoting hatch cover. 2: precipitation sensor. 3: mast. 4: sloped double roof. 5: spirit level. 6: protective handrail. 7: air conditioning unit. 8: puncture-proof pivoting wheel. 9: stabilizing jack. 10: connectors (power, Ethernet). 11: connector roof. 12: forklift rail. 13: removable ABB extension. The screw line on the sloped roof indicates the position of the thermal wall separating spectrometer cabinet and calibration cabinet. The double roof, the hatch cover, and the top of the ABB extension cover are all sloped to avoid pooling or accumulation of liquid and solid water on the enclosure.

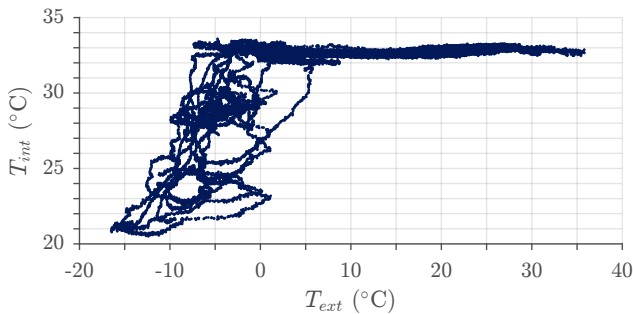

**Figure 8.** Example of interferometer temperature as a function of external temperature (ASSIST-22, 80 days of outdoor operation between August 2023 and February 2024). The interferometer temperature is systematically close to the $32°$ target for external temperatures between 5 and $35°$C. The heater maintains the interferometer temperature above $20°$C for colder external conditions.

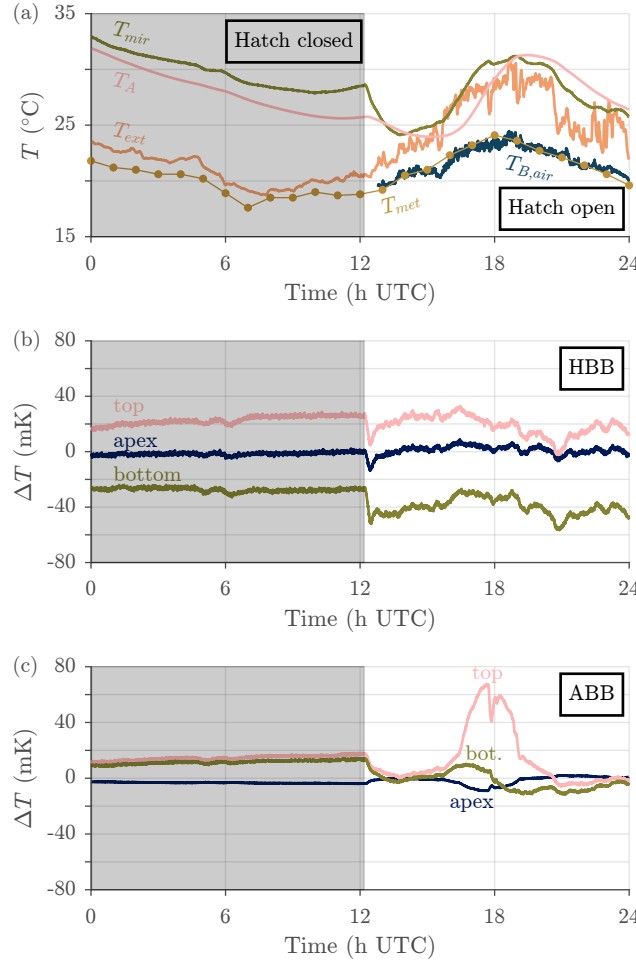

**Figure 9.** Typical calibration module temperatures measured over a full day of uninterrupted outdoor operation (ASSIST-22, June 14 2024). (a) External temperature reported by mast sensor ($T_{ext}$), air temperature reported hourly by a nearby ECCC-MSC meteorological station ($T_{met}$), ABB temperature ($T_A$), and scene mirror ambient temperature ($T_{mir}$). Also shown is the brightness temperature in the opaque 675-680 cm$^{-1}$ region ($T_{B,air}$), derived from the spectrometer sky data and representative of the temperature of the air in the first few meters above the instrument (see Sect. 3.6). The grayed-out area indicates a closed hatch. The meteorological station providing $T_{met}$ is located on the roof of a four-story building approximately 6 km away from the ASSIST, on the other side of the Saint Lawrence River, and its elevation is approximately 35 m higher than that of the ASSIST. (b) HBB temperature readouts relative to the 60°C setpoint. The HBB typically displays a clear vertical temperature gradient. (c) ABB temperature readouts relative to the weighted mean ABB temperature. The top of the ABB cavity was exposed to sunlight between approximately 1600 and 1900 UTC (coordinated universal time), creating a stronger temperature gradient.

## 2.7 Health monitoring system

The ASSIST constantly logs data useful to monitor the instrument status, track changes in the immediate instrument environment, and diagnose potential problems remotely. The most important variables are reported live in the ASSIST graphical user

interface and are also saved in the output files: once per interferometer scan ($\approx 1$ s$^{-1}$ by default) in the raw data files and once per view in the processed data files ($\approx 4$ min$^{-1}$ by default, see Table 6). Included are the data and metadata produced by several ASSIST modules (preamplifiers, cooler, GPS, scene mirror, HBB, ABB, hatch, and mast sensor) as shown in Table 4. In particular, a Bosch BME680 sensor is mounted in a vented cavity underneath the precipitation sensor (Fig. 7) to measure external temperature, pressure, and relative humidity with specified absolute accuracies of 1 K, 60 Pa, and 3 %[3]. A network of 1-Wire

devices (MicroLAN) is also deployed to monitor temperature, voltage, and humidity at key locations inside the interferometer, spectrometer, and calibration modules. The associated "housekeeping" variables are listed in Table 5. The housekeeping temperature sensors have a nominal accuracy of 0.5 K and a readout resolution of 62.5 mK for critical monitoring locations and 500 mK otherwise.

## 3   Signal processing

This second section describes how the digital interferograms are processed by the ASSIST software to yield calibrated radiance spectra (often simply referred to as "radiances") in near real-time. The main processing steps are Fourier transformation and spectral calibration, nonlinearity correction, radiometric calibration, finite field of view correction, resampling, and cropping. All steps are performed independently for the HgCdTe channel and InSb channel from a combination of fixed parameters read from control files (set by LR Tech during instrument production, but user-adjustable) and variables recorded by the health

monitoring system, such as ABB and HBB weighted mean temperatures and interferogram peak values. The forward and reverse scan directions are treated separately until completion of the radiometric calibration, which allows for the electrical and optical dispersion to be treated properly. The resulting radiometrically-calibrated spectra from each scan direction are then averaged together. Finally, the averaged spectrum is further processed to correct for distortions that are independent of scan direction (finite field of view, non-standard spectral grid) and it is cropped to remove the samples that are outside the ASSIST's

sensitivity range.

### 3.1   Spectral representation and calibration

From the $N$ recorded samples of the digital interferogram $I_i[n]$, where the subscript $i$ denotes the scene type ($A$ for "ABB", $H$ for "HBB", $S$ for "sky", $R$ for "reference", with appended $d$ sometimes used to indicate forward $f$ or reverse $r$ scan direction) and the square brackets indicate a discrete argument, the complex spectrum $C_i[k]$ is computed as

$$C_i[k] = e^{j\pi k} \sum_{n=0}^{N-1} I_i[n] e^{-j2\pi nk/N}, \tag{2}$$

where $k = n = 0, 1, ..., N-1$ and where the factor $e^{j\pi k} = (-1)^k$ is added so that an interferogram that is closely symmetrical around $n = N/2$ yields a spectral phase close to zero. The sum in Eq. 2 constitutes the discrete Fourier transform (DFT) of the digital interferogram $I_i[n]$; it is implemented efficiently using the fast Fourier transform (FFT) algorithm. Since the

---

[3]At the time of this writing, there is evidence of daytime solar temperature bias caused by insufficient ventilation of the mast sensor cavity (Fig. 9 and Fig. 16).

**Table 4.** Monitored variables (not exhaustive).

| Module | Variable | Type |
|---|---|---|
| HgCdTe or InSb | Peak-peak amplitude | $N$ (counts) |
| | Index of maximum IGM value | $n$ |
| | Index of minimum IGM value | $n$ |
| Cooler | Setpoint voltage | $V$ (mV) |
| | Voltage (temperature diode) | $V$ (mV) |
| | AC voltage | $V$ (V) |
| | Output frequency | $f$ (Hz) |
| | Controller internal temperature | $T$ (°C) |
| | Lifetime power up count | $N$ |
| GPS | Time | $t$ (UTC) |
| | Latitude | $\phi$ (°) |
| | Longitude | $\lambda$ (°) |
| | Altitude | $H$ (m) |
| Scene mirror | Nominal angle | $\theta$ (°) |
| ABB or HBB | Setpoint temperature | $T$ (°C) |
| | Mean temperature ($T_A$ or $T_H$) | $T$ (°C) |
| | Apex thermistor temperature | $T$ (°C) |
| | Top thermistor temperature | $T$ (°C) |
| | Bottom thermistor temperature | $T$ (°C) |
| Hatch | Open or closed or other status | Integer |
| | Automatic or manual status | Boolean |
| | Clear or precipitation status | Boolean |
| Mast | External temperature ($T_{ext}$) | $T$ (°C) |
| | External pressure | $p$ (kPa) |
| | External humidity | $RH$ (%) |

interferogram is a real signal, its spectrum displays Hermitian symmetry: $C_i[k] = C_i^*[N - k]$, where the star denotes the complex conjugate. Therefore, half of the discrete spectrum can be discarded without loss of information (the last $N/2 - 1$ samples for $N$ even). The OPD axis $x[n]$, associated to the interferogram, and the spectral axis $\upsilon[k]$, associated to its spectrum, are defined as:

$$x[n] = \frac{n - N/2}{\upsilon_s}, \tag{3a}$$

$$\upsilon[k] = \frac{k\upsilon_s}{N}, \tag{3b}$$

**Table 5.** Housekeeping variables.

| Group | Description | Type |
|---|---|---|
| Interferometer | Interf. structure temp. ($T_{int}$) | $T$ (°C) |
| | Interf. air humidity | $RH$ (%) |
| | Secondary input port temp. ($T_R$) | $T$ (°C) |
| Spectrometer | Stirling cooler block temperature | $T$ (°C) |
| | Power supply ambient temperature | $T$ (°C) |
| | Electronics temperature | $T$ (°C) |
| | 24 VDC general | $V$ (V) |
| | 15 VDC FTS board | $V$ (V) |
| | 15 VDC preamplifier board | $V$ (V) |
| | 12 VDC general | $V$ (V) |
| | 5 VDC FTS board | $V$ (V) |
| Calibration | Scene mirror ambient temp. ($T_{mir}$) | $T$ (°C) |
| | Stepper motor temperature | $T$ (°C) |

where $\upsilon_s$ is the sampling wavenumber. These definitions are valid for even values of $N$ and are used for both forward and reverse scan directions. With the ADC triggered once per fringe of the metrology interferogram, $\upsilon_s$ can be related to the vacuum lasing wavenumber $\upsilon_l$ and the laser beam propagation angle $\theta_l$ relative to the interferometer optical axis by (Desbiens et al., 2006b)[4]

$$\upsilon_s = \upsilon_l \cos\theta_l. \tag{4}$$

The spectral calibration, that is, the process by which an apparent wavenumber $\upsilon$ is assigned to a spectral bin $k$, as described by Eq. 3b, therefore only requires knowledge of $\upsilon_s$, the same for both detection channels, and can be highly accurate and repeatable when the two involved parameters ($\upsilon_l$ and $\theta_l$) are stable, which is the case for the ASSIST at a level of a few parts-per-million (ppm) under typical conditions. This straightforward spectral calibration constitutes the so-called Connes' advantage for FTS (Connes, 1958; Buijs and McKinnon, 2009).

The previous definitions, without explicit apodization of the interferogram and without zero-padding, imply that the discrete spectrum $C_i[k]$ is minimally sampled and maximally resolved with associated compromise in dynamic range. For the ASSIST in its default configuration, $N = 2^{15}$ and $\upsilon_s = 15\,798.0$ cm⁻¹ (red He-Ne metrology laser, 632.991 nm in vacuum, with $\theta_l \approx 0$ by design). The natural spectral bin spacing is thus $\upsilon_s/N = 0.48$ cm⁻¹. The implicit truncation of the interferogram, or boxcar

---

[4]Strictly, the refractive index of air (inside the interferometer) at the lasing wavenumber should also be considered, but the infrared interferogram is spatially compressed in almost equal proportion to the metrology interferogram since air is only weakly dispersive from 525 to 15 798 cm⁻¹ ($\approx$ 4 ppm peak index variation) (Edlén, 1966); the small effect or air refraction and associated thermo-optic effect can thus be neglected and all definitions can be made in vacuum.

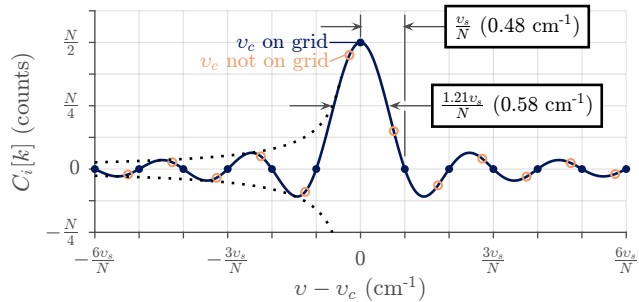

**Figure 10.** Modeled spectrum of the elementary interferogram $I_i[n] = \cos(2\pi v_c x[n])$, where $v_c$ is the "center" modulation wavenumber. The sinc line shape (solid curve, computed as the discrete-time Fourier transform of the digital interferogram, with continuous $v$), $(N/2)\,\mathrm{sinc}\,[(v - v_c)\,N/v_s]$, is caused by the finite OPD span; it is the limiting case of a Dirichlet kernel for large $N$ and $v_c$ far from 0 and $v_s/2$. If $v_c$ is an element of $v[k]$ (Eq. 3b), the samples of the minimally sampled $C_i[k]$ coincide with the peak and the zeros of the underlying continuous line shape function (points). Otherwise, spectral leakage or ringing is visible (circles). The amplitude of this ringing is bounded by $(2\pi|v - v_c|/v_s)^{-1}$ (dotted curve) and is, at most, 21 % of the peak amplitude (Harris, 1978).

apodization, over an OPD spanning $N/v_s = 2.07$ cm is the main factor limiting the spectral resolution. Figure 10 illustrates
the sinc line shape caused by this truncation. This line shape should be understood as the unapodized resolution kernel, or the spectral response to a monochromatic and monoangular (single-field) input radiation, with a full-width at half maximum (FWHM) of $1.21 \times v_s/N = 0.58$ cm[-1] that is slightly larger than the spectral bin spacing[5].

### 3.2   Nonlinearity correction (NLC)

The HgCdTe photoconductor used in the ASSIST is nonlinear: its responsivity decreases with increasing photon irradiance
(Theocharous et al., 2004). The distortion caused by this optical nonlinearity can be corrected to first order in the in-band region by scaling the uncorrected digital interferogram $I_{0,i}[n]$ to compensate the attenuation of the local conversion slope and by adding a quadratic term to compensate the curvature around the point of operation (Taylor, 2014):

$$I_i[n] = (1 + 2a_2 V_{0,i})I_{0,i}[n] + a_2\,(I_{0,i}[n])^2, \tag{5}$$

where $a_2$ is the quadratic nonlinearity coefficient, assumed to be an instrument constant, and $V_{0,i}$ is the modeled DC level
(point of operation) attributed to $I_{0,i}[n]$. The scaling of the interferogram (first term) implies identical scaling of the complex discrete spectrum $C_i[k]$ since the DFT is a linear operator; it constitutes the dominant of the two corrections except at the edges of the in-band region where the transform of the squared interferogram (second term, associated to a spectral self-convolution) typically becomes more important (Fig. 11). Because the InSb detector and its amplification chain are intrinsically linear, the nonlinearity correction (NLC) described here is only applied to HgCdTe interferograms.

---

[5]The FWHM resolution would change with an apodization function different from boxcar while the spectral bin spacing would stay the same.

A model for the DC level is needed primarily because the photocurrent produced by the detector is high-pass-filtered after transimpedance amplification, making $I_{0,i}[n]$ a zero-mean or AC signal[6]. The DC model is similar to that used by the AERI (Knuteson et al., 2004b):

$$V_{0,i} = \eta_m^{-1} \left[ (2 + f_b) (Z_{L,Hd} - Z_{0,Hd} - Z_{L,Rd}) + Z_{0,i} \right]$$
$$\equiv g Z_{0,i} + o_d, \tag{6}$$

where $\eta_m$ is the modulation efficiency (value between 0 and 1, to be characterized for each instrument), $f_b$ is the fraction of background radiation (fixed at 1.0), and $Z$ is the peak value of the uncorrected interferogram ($I_{0,i}[n]$): that of the laboratory HBB view for $Z_{L,Hd}$ (that is, the HBB measurement performed during the laboratory characterization of the nonlinearity coefficients), the most recent HBB view for $Z_{0,Hd}$, the internal reference (self-emission) for $Z_{L,Rd}$, and the current view for $Z_{0,i}$. The DC model is scan-direction specific and represents a linear interpolation with slope $g$ (reciprocal to the modulation efficiency) and offset $o_d$ which is the DC level associated to a zero peak value (balanced interferometer inputs). In general, the peak value $Z$ is not equal to the value of the interferogram at ZPD ($x = 0$ or $n = N/2$) because of dispersion and sampling effects corrected during later radiometric calibration (Sromovsky, 2003). For instance, the peak value of the uncorrected digital interferogram depicted in Fig. 11 is -0.885 MC (megacounts, see Fig. 11) and it is found two samples away from ZPD ($x = -1266$ nm). Finding the minimum or maximum sample of the digital interferogram instead of interpolating around the peak region entails at most a 3 % error given the bandwidth of the HgCdTe channel and the sampling interval.

Figure 11 depicts the impact of the correction described by Eq. 5 on HBB interferograms. Because of the inverting amplifier used after the transimpedance amplifier, measured DC levels are negative for the ASSIST's HgCdTe channel ($a_2$ is thus negative, yielding a positive $2a_2 V_{0,i}$ scaling), HBB interferograms are negative, sky interferograms are typically positive, and ABB interferograms often switch polarity with changes in temperatures. AC peak polarities are the same in the InSb channel. Fitted values of the factor $2a_2 V_{0,Hd}$ (H subscript for the HBB views), which is associated to the highest photon irradiance, are uniformly distributed as 0.10±0.06 for a dozen ASSIST units. The NLC shown in Fig. 11 for ASSIST-22 is close to average with $2a_2 V_{0,Hf} = 0.088$ for the forward scan direction. It is almost identical for the reverse scan direction, with a small difference < 1 % explained by sampling and dispersion effects.

## 3.3 Radiometric calibration

The attribution of physically representative values, in units of radiance, to each sample of the discrete spectrum $C_i[k]$ is performed through two-point complex calibration (Revercomb et al., 1988). Such radiometric calibration is based on the assumption that any complex spectrum $C_i[k]$ can be modeled as $C_i = G_d(L_i + O_d) + M_i$, where $L_i$ is the real incident (one-sided spectral) radiance, $G_d$ is the complex spectral gain ($d$ subscript because it is scan-direction specific), $O_d$ is the complex spectral offset and $M_i$ is the complex zero-mean measurement noise in the spectral domain. The spectral dependence is implicit from this point onward. The "two-point" name refers to two measurements for which the incident radiance $L_i$ is known, here the

---

[6]In a photoconductive detector, the optical DC level is usually dwarfed by the electrical DC level caused by detector biasing. Only the optical component, challenging to accurately measure, matters for NLC.

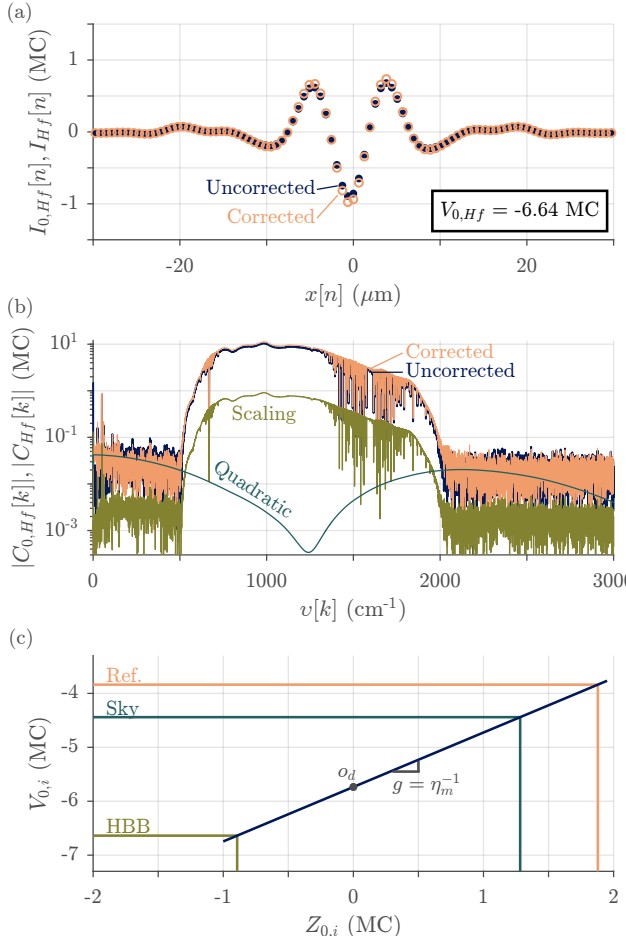

**Figure 11.** Typical impact of the NLC in the HgCdTe channel measured with ASSIST-22. (a) Forward-scan HBB interferograms. The points represent the uncorrected interferogram $I_{0,Hf}[n]$ and the circles the corrected interferogram $I_{Hf}[n]$. Note that the subtle asymmetry of secondary lobes with respect to $x = 0$ is mainly due to dispersion caused by analog filters (high-pass and anti-aliasing). (b) HBB magnitude spectra computed from the same interferograms and contribution of the two correction terms: scaling and quadratic. (c) DC model. The $o_d$ value changes slowly over time, as captured by the $(Z_{L,Hd} - Z_{0,Hd})$ component of Eq. 6, when the interferometer temperature changes. Here, $a_2 = $ -6.62 $\times$ $10^{-3}$ MC$^{-1}$, $\eta_m = 0.99$, $Z_{L,Hf} = $ -0.907 MC, $Z_{0,Hf} = $ -0.885 MC, $Z_{L,Rf} = $ 1.879 MC, and $Z_{0,Sf} = $ 1.273 MC. MC = $10^6$ counts, where a count is defined as the 16-bit bipolar ADC level multiplied by 128 and divided by the programmable analog gain (1,2,4,..., 128). For the measurements shown here, the programmable gain is set to 2 and ADC saturation therefore occurs at $\pm 2^{15} \times 128/2 = \pm 2.1$ MC.

HBB and ABB measurements, together allowing the estimation of the unknown spectral gain and offset through

$$\widehat{G}_d = \frac{C_{Hd} - C_{Ad}}{L_H - L_A}, \tag{7a}$$

$$\widehat{O}_d = \frac{L_H C_{Ad} - L_A C_{Hd}}{C_{Hd} - C_{Ad}}, \tag{7b}$$

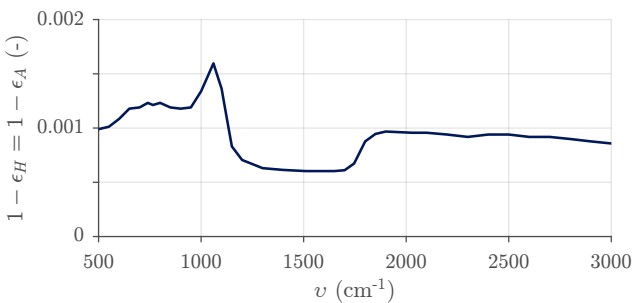

**Figure 12.** Effective reflectivity model used in the blackbody cavity radiance model. The blackbody emissivity is complementary.

where the hat symbol denotes an estimated value and where the two blackbody cavity radiances are modeled as (ignoring radiative transfer through the intermediate gas)

$$L_H = \epsilon_H L_P(T_H) + (1 - \epsilon_H) L_P(T_r), \tag{8a}$$

$$L_A = \epsilon_A L_P(T_A) + (1 - \epsilon_A) L_P(T_r), \tag{8b}$$

with $\epsilon_i$ as the effective blackbody spectral emissivity, $T_i$ as the effective blackbody temperature, $T_r$ as the "reflected" temperature, that is, the radiative temperature of the environmental emission reflected by the imperfect blackbody cavity into the spectrometer's FOV, and where $L_P(T)$ is the wavenumber Planck function defined as

$$L_P(T) = \frac{2hc^2 v^3}{e^{hcv/k_B T} - 1}, \tag{9}$$

with $h$ as the Planck constant, $c$ as the speed of light in vacuum, and $k_B$ as the Boltzmann constant.

Since the ASSIST's blackbody cavities are almost exact replicas of the AERI's (in terms of geometry and internal surface treatment), the model used for $\epsilon_H$ and $\epsilon_A$ is very similar to that used for the AERI (Fig. 12): the cavity factor is assumed to be 39, and the modeled effective emissivity is higher than 0.9984 in the spectral range of interest. Moreover, the effective emissivity is supposed to be temperature-independent in the range of operation and $\epsilon_H = \epsilon_A$ is therefore considered to be valid since the two cavities are nominally identical. The effective temperature $T_i$ is the weighted mean of readouts from the three calibrated thermistors (Fig. 5), as described previously. The reflected temperature $T_r$ can also be taken as the readout from several onboard sensors, but by default it is the calibration module temperature measured near the scene mirror ($T_{mir}$).

Although the estimated gain and offset, $\widehat{G}_d$ and $\widehat{O}_d$, are only intermediate products of the radiometric calibration, they do provide valuable information about the status and performance of the spectroradiometer. The magnitude of $\widehat{G}_d$ is related to the spectral responsivity from calibration plane (the plane where the aperture of the blackbody cavities are located) to detector; it depends on geometric étendue, transmission of optics and intermediate gas, photodetector responsivity, transimpedance amplification, and preamplifier frequency response. The phase of $\widehat{G}_d$ is related to electrical dispersion, residual optical dispersion, and the position of the sampling grid relative to the analog interferogram, which depends on the scan direction. Figure 13 shows the typical responsivity (magnitude of the complex gain) in the 500 to 3000 cm$^{-1}$ range for the forward scan direction;

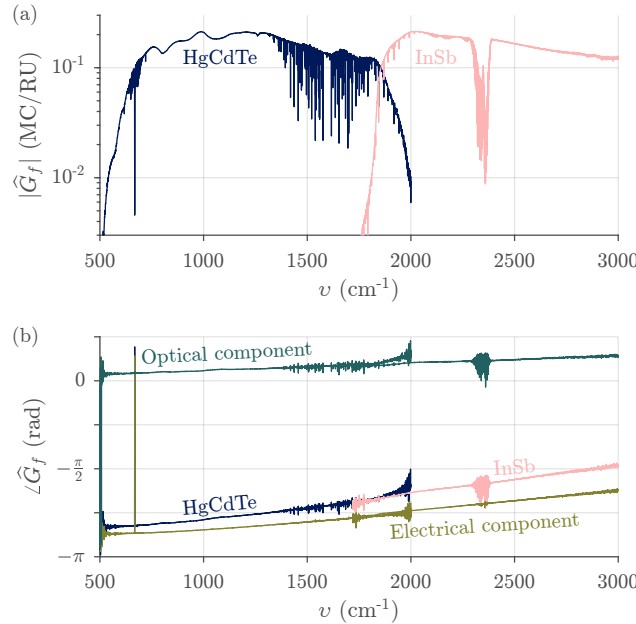

**Figure 13.** Typical complex spectral gain measured with ASSIST-22 for a forward scan. (a) Magnitude (also called responsivity). (b) Phase, total and decomposed. The "electrical" component of the phase is common to both forward and reverse scan directions. The "optical" component of the phase switches polarity with scan direction. These components can be extracted by summing and subtracting the total phases for the forward and reverse scan directions. Absorption lines of internal $H_2O$, $CO_2$, and CO (molecules present between calibration plane and detector) are also visible. RU = mW $(m^2\ sr\ cm^{-1})^{-1}$.

the responsivity for the reverse scan direction is the same within the measurement noise. The complex spectral offset $\widehat{O}_d$ is associated to the self-emission of the instrument, that is, the measured radiance for a zero scene radiance ($L_i = 0$). It is generally complex since it includes contribution from the input port (fore optics), the beamsplitter, and the reference port that all display different phases (Kleinert and Trieschmann, 2007; Runge et al., 2021). For the ASSIST, the contribution from the reference

port is the most important and $\widehat{O}_d$ is thus dominated by a real and negative component that resembles a Planck curve. This curve closely follows the slowly fluctuating temperature $T_R$ of the blackened surface used in the reference port, normally close to 32°C under steady-state operation (Fig. 14).

Both gain and offset can and do change over time, following diurnal changes in temperatures, instrument warm-up and aging, fluctuations in internal gas composition, deposition of dust on the scene mirror, etc. Frequent calibration is therefore required

to reach an adequate radiometric accuracy. To this end, the ASSIST's scene mirror operates on a repeating view schedule of the following general form:

$$\overbrace{AH\,(SS...S)}^{c_1}\,H\,A\,\underbrace{(SS...S)}_{c_2}\,\overbrace{AH\,(SS...S)}^{c_3}\,H\,A\,..., \tag{10}$$

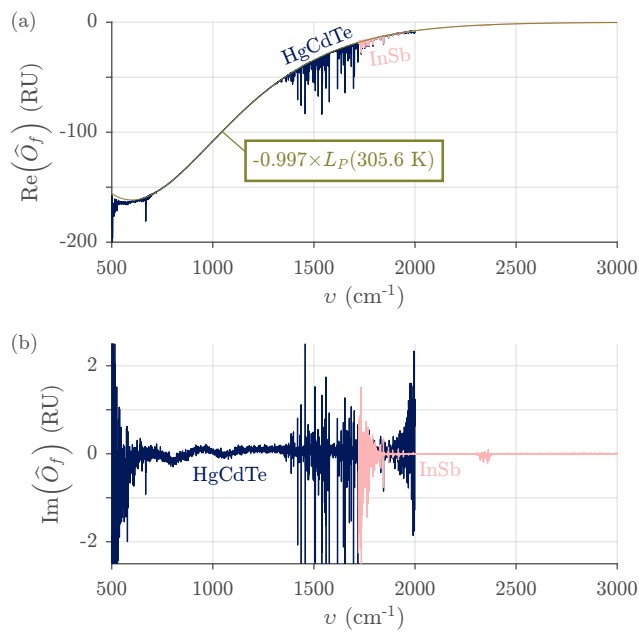

**Figure 14.** Typical complex spectral offset measured with ASSIST-22 for a forward scan. (a) Real part. The smooth curve is a fitted graybody radiance (the simultaneous housekeeping sensor readout was $T_R = 304.9$ K). (b) Imaginary part. Absorption lines of internal $H_2O$ and $CO_2$ are visible in both cases. RU = mW $(m^2$ sr $cm^{-1})^{-1}$.

where $c$ refers to a calibration cycle or calibration run for which all acquired data are batched for processing, independently from other cycles (incomplete calibration cycles are not processed). Each calibration cycle therefore contains an adjustable number of sky views $S$ bracketed by views of the two onboard blackbodies: either $AH - HA$ or $HA - AH$. The number of scans per view is also adjustable, but forced the same for all views (12 by default: six forward scans and six reverse scans). During processing, the forward scan interferograms from the two detection channels (corrected for nonlinearity in the HgCdTe channel) are coadded to yield a single representative forward scan interferogram $I_{if}[n]$ per view and per channel; the same is done for reverse scan interferograms. The default values of all adjustable parameters are presented in Table 6. Accounting for scene mirror movement and intentional pauses, a calibration cycles completes in approximately 160 s under the normal conditions of Table 6, and there are 653±1 calibration cycles (3918±6 sky views) in a full day of uninterrupted operation (or 132 s between equivalent sky views of adjacent calibration cycles). Regardless of how the view schedule is adjusted, the calibration interferograms bracketing the sky views are linearly interpolated to the central time of each sky view to remove, to first order, potential drifts in instrument gain, in instrument offset, and in calibration radiances which normally happen slowly relative to the duration of a calibration cycle given the thermal masses involved (see Figs. 9 and 16). After DFT and using the effective cavity temperatures associated to that time, this temporal interpolation process provides refined estimates of the gain and offset (Eq. 7) valid for a specific sky view and scan direction and can thus be renamed $\widehat{G}_{Sd}$ and $\widehat{O}_{Sd}$.

**Table 6.** Scene mirror view schedule for ASSIST-22.

| Index | Scene | Angle | Views (rep.) | Scans | Dwell time |
|-------|-------|--------|--------------|--------------|------------|
| 1 | $A$ | 62.0° | 1 | 12 | 12.6 s |
| 2 | $H$ | 300.3° | 1 | 12 | 12.6 s |
| 3 | $S$ | 0.0° | 6 | $6 \times 12$ | 86.3 s |
| 4 | $H$ | 300.3° | 1 | 12 | 12.6 s |
| 5 | $A$ | 62.0° | 1 | 12 | 12.6 s |
| 6 | $S$ | 0.0° | 6 | $6 \times 12$ | 86.3 s |

Finally, the radiance $L_{Sd}$ for each channel, each scan direction, and each sky view is estimated from the refined gain and offset estimates through

$$\widehat{L}_{Sd} = \mathrm{Re}\left( \frac{C_{Sd}}{\widehat{G}_{Sd}} - \widehat{O}_{Sd} \right), \tag{11}$$

where $\mathrm{Re}$ refers to the real part of the complex argument. Though the form is different, the previous definition is entirely equivalent to Eq. 4 in Knuteson et al. (2004b) and can be shown to describe a linear interpolation or extrapolation, on a wavenumber-by-wavenumber basis and in the complex domain, of a given spectrum with respect to the two calibration spectra (refined through temporal interpolation). Obviously, the estimated radiance $\widehat{L}_{Sd}$ is meaningful only in spectral regions where $\widehat{G}_{Sd}$ is sufficiently high and where the signal-to-noise ratio (SNR) is significantly above unity (or the SNR on the responsivity is above 3.3, as explained in Rowe et al. (2011b)); though this depends on the dwell time per view and the calibration temperatures, sensitivity ranges are typically 500 to 2050 cm$^{-1}$ for the HgCdTe channel and 1700 to 3650 cm$^{-1}$ for the InSb channel, with some subranges occasionally degraded by strong absorption lines of internal $H_2O$ and $CO_2$ (Rowe et al., 2011a). Figure 15 shows the calibrated radiance $\widehat{L}_{Sd}$ for a typical clear sky view. The spectral regions where the radiance is well below the 281 K Planck curve (green) correspond to high atmospheric transparency. Emission lines from tropospheric gaseous species ($H_2O$, $CO_2$, $N_2O$, $CH_4$, $O_3$, and CO) are visible. Some lines of HDO can also be distinguished barely above the measurement noise in the 2500 to 2900 cm$^{-1}$ range.

The imaginary part of the calibrated complex spectrum also provides important information about the level of merit of the calibration process and is defined here as $\widehat{D}_{Sd}$:

$$\widehat{D}_{Sd} = \mathrm{Im}\left( \frac{C_{Sd}}{\widehat{G}_{Sd}} - \widehat{O}_{Sd} \right), \tag{12}$$

where $\mathrm{Im}$ refers to the imaginary part of the complex argument. A perfect radiometric calibration only leaves noise in $\widehat{D}_{Sd}$, and this noise displays the same statistics as the noise found in $\widehat{L}_{Sd}$ if the additive noise in the interferogram is a wide-sense-stationary random process (Sromovsky, 2003). Any non-vanishing signal found in $\widehat{D}_{Sd}$ can indicate a problem with calibration or significant instrumental drift that is not properly corrected by linear interpolation of the bracketing calibration views.

At this stage of signal processing, the instrument's phase signature for both scan directions is supposed to be adequately corrected and the (real) radiances $\widehat{L}_{Sd}$ associated to forward and reverse scan directions are averaged to yield a single radiance

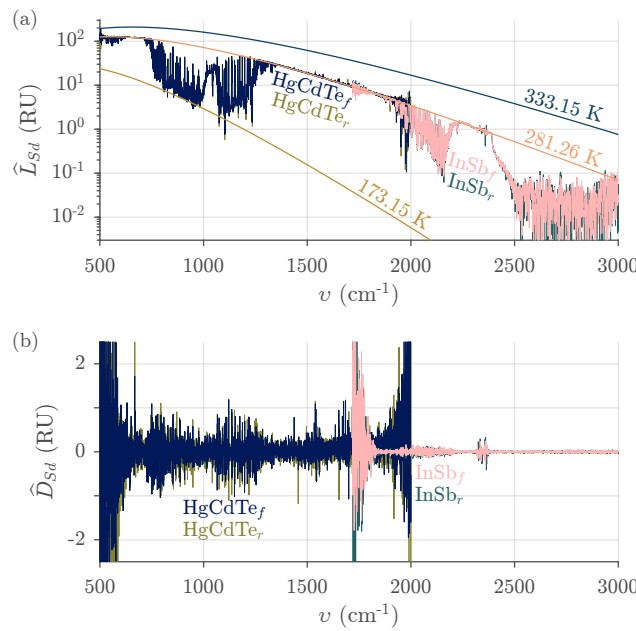

**Figure 15.** Typical calibrated complex spectra measured with ASSIST-22 for the two scan directions. (a) Real part. The forward scan direction spectrum is traced on top of the reverse scan direction spectrum. Planck curves at different temperatures are also illustrated: 60°C (HBB), -100°C, and 281.26 K (representative of the temperature of the air in the first few meters above the instrument, the simultaneous mast sensor readout was $T_{ext} = 281.3$ K). (b) Imaginary part. RU = mW (m$^2$ sr cm$^{-1}$)$^{-1}$.

estimate for each sky view and each channel: $\widehat{L}_S = (\widehat{L}_{Sf} + \widehat{L}_{Sr})/2$. The same is done for the imaginary components and the interpolated responsivities. Figure 16 illustrates how the radiometric calibration parameters typically evolve over a full day of operation. Despite a 14°C variation in external air temperature and a 19°C variation in ABB temperature during that day, 445 the responsivity extracted through radiometric calibration remains stable, implying accurate modeling of effective temperatures and emissivities. Moreover, the HBB and reference port temperatures do not fluctuate significantly around their setpoint values.

### 3.4 Finite field of view (FFOV) correction

A monochromatic ($\upsilon_c$) radiation source filling the instrument's finite FOV does not yield a pure cosine interferogram oscillating at $\upsilon_c$ (Genest and Tremblay, 1999). Since the modulation frequency is proportional to the cosine of a ray's propagation 450 angle inside the interferometer, the spectrum produced by the instrument is distorted: it shows a response distributed between $\upsilon_c \cos\theta_{max}$ and $\upsilon_c \cos\theta_{min}$ instead of a single infinitesimal component at $\upsilon_c$, where $\theta_{max}$ and $\theta_{min}$ are the maximum and minimum allowed propagation angles relative to the optical axis[7]. The spectral resolution is thus limited further than predicted from the interferogram's implicit truncation alone (finite maximum OPD or boxcar apodization).

---

[7]Unless an entrance telescope or entrance lens is used, propagation angles through the interferometer are equal to field angles. $\theta_{min}$ is 0 in the case of propagation along the optical axis, which is allowed for the ASSIST.

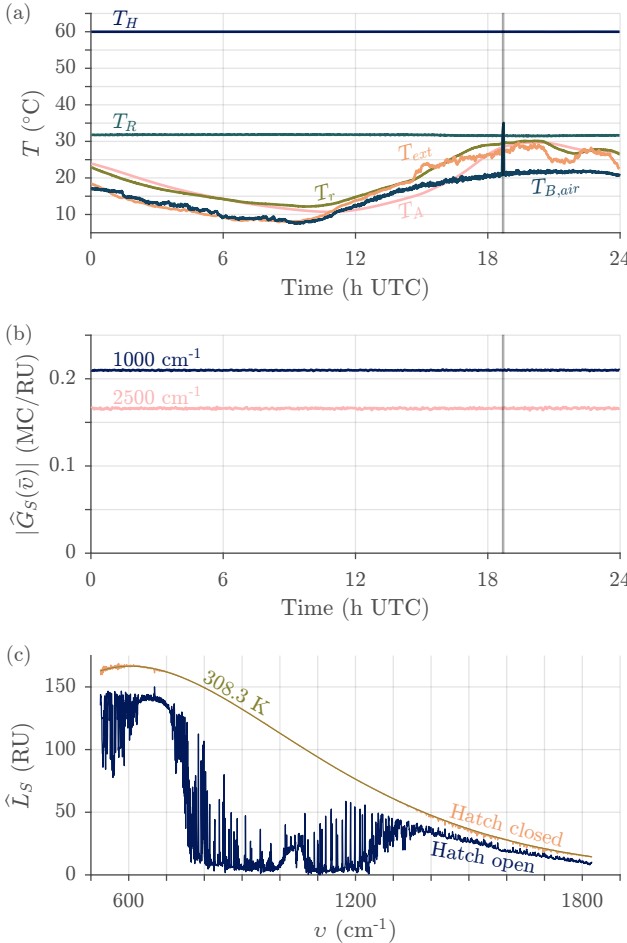

**Figure 16.** (a) Typical temperatures involved in radiometric calibration measured over a full day of uninterrupted outdoor operation (ASSIST-22, June 16 2024): HBB temperature ($T_H$), ABB temperature ($T_A$, the peak temperature variation is 165 mK over 132 s), reference port temperature ($T_R$), and reflected temperature ($T_r$). Also shown are the external temperature ($T_{ext}$) and the brightness temperature in the opaque 675-680 cm$^{-1}$ region ($T_{B,air}$), representative of the temperature of the air in the first few meters above the instrument. (b) Sampled responsivity associated to a forward scan near $\bar{v} = 1000$ cm$^{-1}$ (HgCdTe channel) and near $\bar{v} = 2500$ cm$^{-1}$ (InSb channel) for the same period. The hatch closed automatically due to precipitation for a brief period around 1840 UTC (shaded gray). (c) Transition from hatch cover's inner surface radiance (1843 UTC) to sky radiance after the precipitation event (1844 UTC) in the HgCdTe channel. The smooth curve is a fitted Planck curve representative of the hatch cover's inner surface temperature. RU = mW (m$^2$ sr cm$^{-1}$)$^{-1}$.

This finite field of view (FFOV) distortion, often described as aperture broadening of the spectrum or self-apodization of
the corresponding interferogram, can be described more generally by an instrument line shape (ILS), $F(v, v_0)$, relating an
arbitrary true spectrum $L_0(v_0)$ (not monochromatic) to the distorted spectrum $L_F(v)$ through a Fredholm equation of the first

kind:

$$L_F(v) = \int\limits_{0}^{\infty} L_0(v_0)F(v,v_0)dv_0,$$ (13)

where $v_0$ is the true wavenumber and $v$ the apparent wavenumber measured by the spectrometer (continuous in this case).
This equation is not in general equivalent to a convolution[8]. The FFOV ILS $F(v,v_0)$ corresponds to the spectrum measured by the FTS for a monochromatic true spectrum, normalized by the power received at the true wavenumber. It can be linked to the radiant intensity inside the interferometer (Desbiens et al., 2002), which depends on the optical design and alignment and generally varies with true wavenumber $v_0$. It also depends on the radiometric characteristics of the source (instrument line shape is a misnomer in this case), though it is modeled here under the simplifying assumption of a uniform Lambertian scene. Therefore, the FFOV ILS model does not properly capture the distortion associated to nonuniform scenes such as a clear sky behind scattered clouds. Both the FFOV ILS and the "truncation" ILS (sinc lineshape depicted in Fig. 10) can be combined in a single "total" ILS kernel if desired, but only the FFOV ILS can be corrected through signal processing, leaving the fundamental sinc lineshape of the truncation ILS.

A common FFOV ILS model is that associated to an achromatic, radially symmetrical, and uniformly irradiated field stop defining a conical FOV with half-angle $\theta_{max} = b$. In the limits of the paraxial approximation, it yields a boxcar lineshape represented as (Vanasse and Sakai, 1967; Genest and Tremblay, 1999)

$$F(v,v_0) = \begin{cases} \frac{1}{v_0(1-\cos b)} & \text{if } v_0\cos b < v < v_0, \\ 0 & \text{otherwise.} \end{cases}$$ (14)

The top panel of Fig. 17 shows this boxcar lineshape on a normalized spectral axis for the case $b = 23$ mrad. Note that the boxcar width is linearly proportional to $v_0$, yielding increasingly severe broadening with increasing wavenumber. Furthermore, since the boxcar's mean is located at $v = v_0(1+\cos b)/2 < v_0$, the broadened spectrum is shifted to the left of the true spectrum (Fig. 17, bottom panel). However, because the relative shift is the same at all true wavenumbers, the measured spectrum simply appears linearly (uniformly) compressed in $v$ relative to the true spectrum. This ILS model is used for the AERI (Knuteson et al., 2004b), with the value of $b$ being either 15, 16, or 23 mrad, depending on the AERI version (Knuteson et al., 2004a; Mariani et al., 2012).

In the ASSIST, the field stop is the clear diameter of the ZnSe lens that has a low f-number (Fig. 3) and the simple boxcar model of Eq. 14 does not fully capture the FFOV distortion caused by its chromatic aberration. Ray tracing shows that on-axis rays are over-represented, that the distribution of allowed propagation angles tapers off progressively instead of abruptly, and that this distribution is chromatic because of dispersion inside the ZnSe material (especially close to the longwave cutoff where the FOV becomes choked). The mean of the FFOV ILS varies with normalized wavenumber (Fig. 17, middle); the corollary is that the relative spectral shift varies with wavenumber, causing a non-uniform compression of the spectral axis. Furthermore, the shape of the distribution is sensitive to detector focus and differs in both detection channels. An example is shown in the

---

[8]The Fredholm equation does however simplify to a simple convolution for the spectrally constant truncation ILS (Desbiens et al., 2006b).

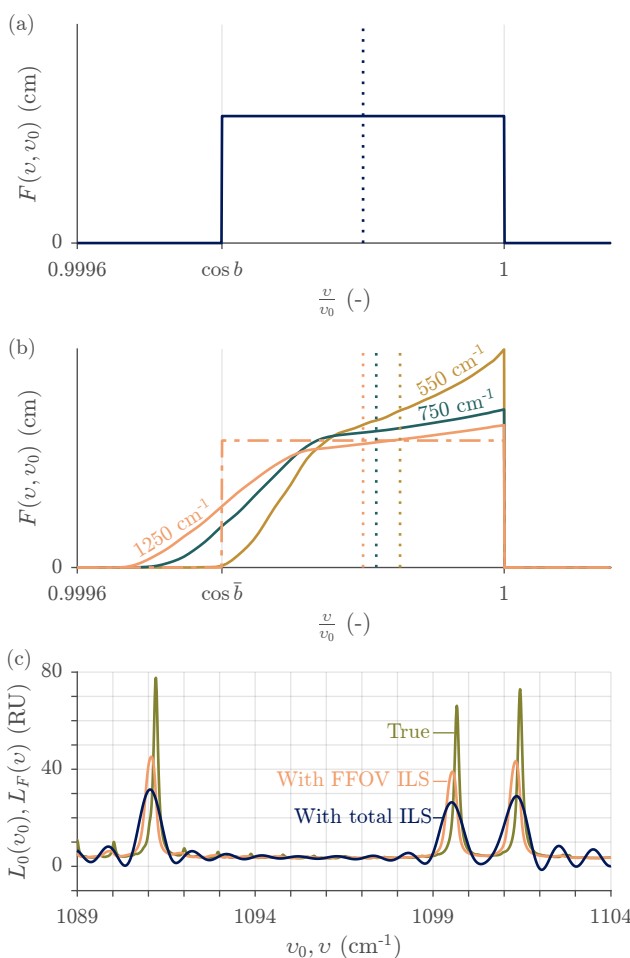

**Figure 17.** FFOV instrument line shape model for the HgCdTe channel. (a) Simple boxcar model for $b = 23$ mrad. (b) Representative ray tracing model for the ASSIST's spectrometer. The three solid curves are the ray tracing model at 550, 750, and 1250 cm$^{-1}$ for a specific detector focus that is close to optimal. The dotted vertical lines are the associated distribution means and the dot-dashed curve is the effective boxcar FFOV ILS at 1250 cm$^{-1}$ (same mean), with $\bar{b} = 23$ mrad like in the top panel. (c) Example of true radiance and associated FFOV-distorted radiance showing both line shift and broadening. The FFOV ILS is the ASSIST model at 1250 cm$^{-1}$. Also shown is the radiance after interferogram truncation (finite maximum OPD), with total ILS (both FFOV ILS and truncation ILS applied). RU = mW (m$^2$ sr cm$^{-1}$)$^{-1}$.

middle panel of Fig. 17 for a specific value of the detector focus in the HgCdTe channel. While such a complicated FFOV ILS could be, in theory, corrected through matrix inversion (Desbiens et al., 2006a), its chromatic nature would make it extremely taxing from a computational standpoint. For this reason, an effective boxcar width $\bar{b}$ is calculated in a specific wavenumber region for each detection channel, as exemplified by the purple dot-dashed curve of Fig. 17. The FFOV correction is then applied without considering the known chromatism and asymmetric shape, making it more performant in the spectral region from which $\bar{b}$ is estimated. There used to be a large spread in the values of $\bar{b}$, from 20 to 30 mrad near 1250 cm$^{-1}$ for different

ASSIST instruments, but the detector alignment procedure was refined in 2023 to tighten the distribution to the 21 to 24 mrad range, primarily by performing rapid characterization on the production line to enhance feedback during the fine-tuning of the detector focus.

The FFOV correction is split in two independent steps in the ASSIST's processing: shift correction and broadening correction. The spectral shift is first corrected by redefining the spectral axis of Eq. 3b using the compensated sampling wavenumber $v'_s$ instead of the true, on-axis sampling wavenumber $v_s$ (Eq. 4). For an ILS with effective boxcar-width $\bar{b}$, both sampling wavenumbers are related through

$$v'_s = \left( \frac{2}{1 + \cos \bar{b}} \right) v_s \approx \left( 1 + \frac{\bar{b}^2}{4} \right) v_s, \tag{15}$$

where the final approximation is valid for $\bar{b} \ll 1$ rad. This redefinition implies the stretching of the spectral axis (from $v$ to $v'$) by a factor that depends on the FFOV ILS only, independent from the metrology laser parameters of Eq. 4 but specific to each detection channel. The spectral samples of the sky radiance $\widehat{L}_S$ are not modified by the stretching operation and the reciprocal factor is used to compress the OPD axis (from $x$ to $x'$). Obviously, this first correction is performed on spectra associated to truncated interferograms, which are affected by both FFOV ILS and truncation ILS. However, the reasoning remains the same since the sinc ILS kernel is symmetric and does not shift spectral features. The FFOV broadening is then corrected using a method similar to that described in Knuteson et al. (2004b), yielding the modified sky radiance $\widehat{L}'_S$:

$$\widehat{L}'_S = \widehat{L}_S + \frac{\left[ 2\pi(\bar{b}^2/4) \right]^2}{3!} \mathcal{F} \left[ x'^2 \mathcal{F}^{-1} \left( \tilde{v}^2 \widehat{L}_S \right) \right], \tag{16a}$$

$$\tilde{v} = \begin{cases} \frac{k v'_s}{N} & k = 0, 1, ..., N/2, \\ \frac{(N-k) v'_s}{N} & \text{otherwise}, \end{cases} \tag{16b}$$

where the $\mathcal{F}$ and $\mathcal{F}^{-1}$ operators describe the modified DFT (Eq. 2) and inverse DFT, respectively:

$$\mathcal{F}(\xi[n]) = e^{j\pi k} \sum_{n=0}^{N-1} \xi[n] e^{-j2\pi nk/N}, \tag{17a}$$

$$\mathcal{F}^{-1}(\xi[k]) = \frac{1}{N} \sum_{k=0}^{N-1} e^{-j\pi k} \xi[k] e^{j2\pi nk/N}. \tag{17b}$$

This is a first-order correction only, and extra terms could be included for improved accuracy. Outside the sensitivity range ("out-of-band") and under strong internal absorption lines, the radiance estimate $\widehat{L}_S$ displays strong noise because of the division by a low or purely random spectral gain during radiometric calibration (Eq. 11) (Rowe et al., 2011b). It is therefore good practice to either replace out-of-band spectral samples by an adequate Planck curve or null-out those spectral samples, with a smooth transition to in-band samples in any case, before applying the correction described by Eq. 16a. Figure 18 depicts the typical effect of FFOV correction over atmospheric water vapor lines visible in the HgCdTe channel. The FFOV broadening correction is also performed on the imaginary component $\widehat{D}_S$, yielding $\widehat{D}'_S$.

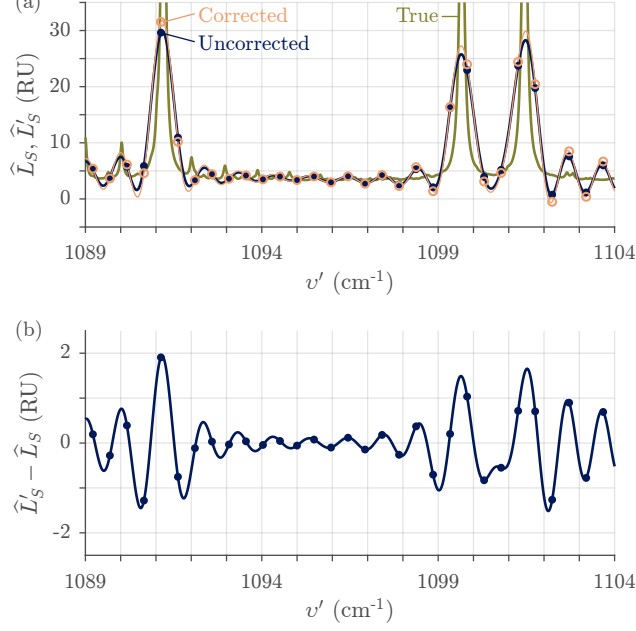

**Figure 18.** Typical impact of the FFOV broadening correction measured with ASSIST-22 (with $\bar{b} = 27.0$ mrad in the HgCdTe channel after a +182 ppm stretching operation). (a) Radiances. The points represent the uncorrected radiance and the circles the corrected radiance, in both cases over the compensated (stretched) spectral axis. The solid curves are computed through zero-padding (spectral sinc interpolation over the DFT samples) to aid the eye. The true radiance (no ILS) is replicated from Fig. 17. (b) Correction detail. RU = mW (m$^2$ sr cm$^{-1}$)$^{-1}$.

## 3.5 Resampling

The values of the sampling wavenumber $\upsilon_s$ and the compensated sampling wavenumber $\upsilon_s'$ are different for different ASSIST instruments (distribution of $\upsilon_s = 15\ 797.2 \pm 0.4$ cm$^{-1}$, $\pm \sigma$, for a dozen ASSIST instruments; $\upsilon_s' = 15\ 799.7 \pm 0.5$ cm$^{-1}$ is the corresponding distribution of the compensated sampling wavenumber in the HgCdTe channel (Eq. 15) and is similar in the InSb channel), leading to variable spectral grids. This complicates data intercomparison and comparison with radiative transfer models. To alleviate this issue, standard grids are defined as

$$x''[n] = \frac{n - N/2}{\upsilon_s''}, \tag{18a}$$

$$\upsilon''[k] = \frac{k\upsilon_s''}{N}, \tag{18b}$$

where the standard sampling wavenumber $\upsilon_s''$ is set to exactly 15 799 cm$^{-1}$, also facilitating comparisons with the AERI (Knuteson et al., 2004b) and defining the spectral bin spacing as exactly (15 799 / 32 768) cm$^{-1}$. FFOV-corrected sky radiances $\widehat{L}_S'[k]$ are resampled from their compensated spectral grid $\upsilon'[k]$ (specific to detection channel and instrument) to the standard spectral grid $\upsilon''[k]$, yielding the final spectrum $\widehat{L}_S''[k]$, using a method that is different from that used for the AERI: instead of a linear interpolation over a spectrum densified through zero-padding, a spline interpolation is performed in the interferogram

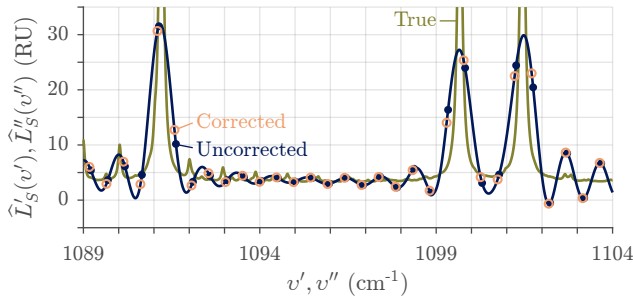

**Figure 19.** Typical impact of the resampling correction measured with ASSIST-22 (with $v'_s = 15\,799.60$ cm$^{-1}$ in the HgCdTe channel, for a -38 ppm resampling). The points represent the uncorrected radiance over the specific axis and the circles the corrected radiance over the standard spectral axis. The solid curve is computed through zero-padding (spectral sinc interpolation over the DFT samples) to aid the eye, and passes through both discrete spectra. The true radiance (no ILS) is replicated from Fig. 17. RU = mW (m$^2$ sr cm$^{-1}$)$^{-1}$.

domain (from $x'$ to $x''$) where the signal varies more slowly and smoothly given the small ratio of detection bandwidth to sampling wavenumber (see interferograms in Fig. 11). Densification is not required for adequate numerical accuracy, but

minor extrapolation is needed given that $v''_s < v'_s$ is usually true. After Fourier transformation, this interferogram-domain interpolation yields a resampled spectrum (from $v'$ to $v''$) that is similar to that obtained through spectral-domain interpolation, but at a reduced computational cost. Figure 19 displays the outcome of this resampling procedure near atmospheric water vapor lines. The resampling is also performed on the imaginary component $\widehat{D}'_S$, yielding $\widehat{D}''_S$, and on the responsivity $|\widehat{G}_S|$ (no FFOV correction), yielding $|\widehat{G}''_S|$.

**3.6   Data products**

Resampled spectra from the two detection channels are finally cropped to remove the out-of-band spectral samples and minimize the size of output files. The cropping limits have default values of 525-1825 cm$^{-1}$ for the HgCdTe channel and 1720-3300 cm$^{-1}$ for the InSb channel. The 1720-1825 cm$^{-1}$ overlap region is useful to assess the effectiveness of the NLC in the HgCdTe channel. Figures 20 and 21 show typical cropped final spectra for both detection channels along with examples of the

magnitude of the main corrections (NLC and FFOV correction). From Eq. 16a, the magnitude of the FFOV correction scales as $\bar{b}^4$ and is therefore unusually large for ASSIST-22 ($\bar{b} = 27.0$ mrad in the HgCdTe channel, $\bar{b} = 26.8$ mrad in the InSb channel) whose results are shown in those two figures.

The processing steps described up to this point, with general properties summarized in Table 7, are executed in near real-time when a valid calibration cycle is completed. For each detection channel, cropped resampled radiances ($\widehat{L}''_S$), cropped resampled

imaginary radiances ($\widehat{D}''_S$), and cropped resampled responsivities ($|\widehat{G}''_S|$) associated to a calibration cycle's sky views (six by default) are appended to single-precision floating-point matrices in NetCDF-3 files created at the beginning of the UTC day. Data produced by the GPS, the hatch, and the mast sensor along with calibration temperatures and scene mirror nominal angles for each sky view are also appended to those two files after each calibration cycle. The final file size, for a complete day of

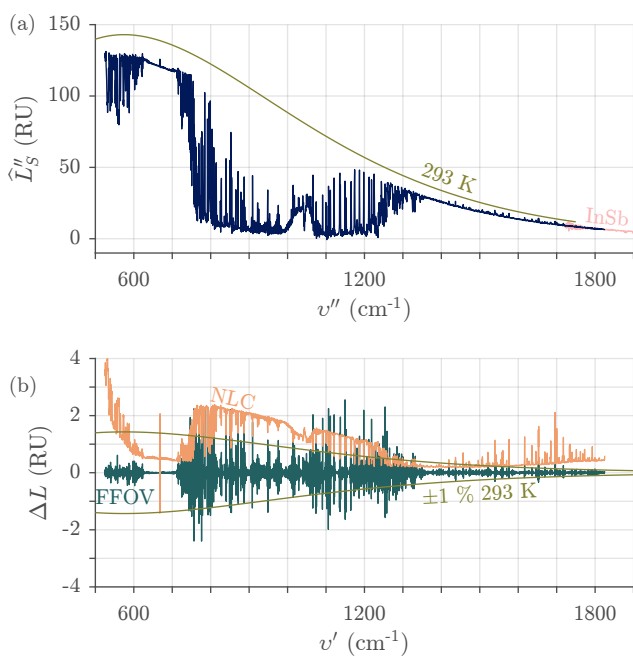

**Figure 20.** (a) Typical final radiance in the HgCdTe channel and for a clear sky view, measured with ASSIST-22 (medium nonlinearity, high FFOV-distortion). The curve above 1720 cm$^{-1}$ is the tail of the measurement in the InSb channel. The smooth curve is a 293 K Planck curve. (b) Associated main corrections, nonlinearity (corrected minus uncorrected) and finite field of view. The smooth curve represents $\pm 1$ % of the 293 K Planck curve. RU = mW (m$^2$ sr cm$^{-1}$)$^{-1}$.

**Table 7.** Processing parameters.

| Parameter | Nominal value |
|---|---|
| Spectral bin spacing | 15 799 / $2^{15} \approx 0.48$ cm$^{-1}$ |
| Apodization function | Boxcar (implicit) |
| FWHM spectral resolution | 0.58 cm$^{-1}$ |
| Nonlinearity presets (HgCdTe) | $a_2, \eta_m, Z_{L,Hd}, Z_{L,Rd}$ |
| BB temperature weights | 0.8-0.1-0.1 (apex-top-bottom) |
| Reflected temperature ($T_r$) | Scene mirror ($T_{mir}$) |
| Calibration bracketing | Linear interpolation |
| FFOV ILS model | Boxcar (for correction) |
| FFOV corr. preset (HgCdTe / InSb) | $\bar{b}$ |
| Resampling preset (HgCdTe / InSb) | $v'_s, v''_s = 15\ 799$ cm$^{-1}$ |
| Cropping ranges (HgCdTe / InSb) | 525-1825 / 1720-3300 cm$^{-1}$ |

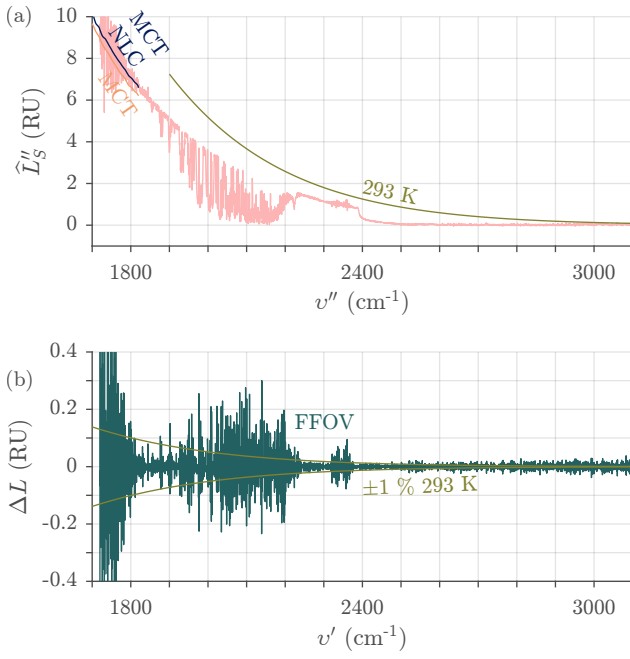

**Figure 21.** (a) Typical final radiance in the InSb channel and for a clear sky view, measured with ASSIST-22 (high FFOV-distortion). The curves below 1825 cm$^{-1}$ are the tails of the measurements in the HgCdTe channel, with and without NLC, smoothed in both cases to better expose the overlap with the InSb signal. The smooth curve is a 293 K Planck curve. (b) Associated finite field of view correction. The smooth curve represents $\pm 1$ % of the 293 K Planck curve. RU = mW (m$^2$ sr cm$^{-1}$)$^{-1}$.

uninterrupted operation, is typically 127 MB for the HgCdTe channel and 154 MB for the InSb channel; most of that size is
555 occupied by the three spectro-temporal matrices (in the HgCdTe (InSb) channel: 653 calibration cycles per day $\times$ 6 sky views per calibration cycle $\times$ 3 saved cropped spectra per view $\times$ 2697 (3278) real-valued samples per cropped spectrum $\times$ 4 bytes per real-valued sample = 127 MB (154 MB) per day). An option is available to save the data collected during the $A$ and $H$ calibration views of each calibration cycle, including the calibrated radiances, for an approximate $8/6$ inflation in file size (Eq. 10).

A separate "summary" file is created at the beginning of each UTC day to store the metadata (settings, NLC and FFOV correction parameters, etc.), the monitored variables (Table 4), the housekeeping variables (Table 5), and derived quality metrics such as the sampled responsivity at 1000 and 2500 cm$^{-1}$ (Fig. 16) as well as calculated brightness temperatures, mean radiances, and standard deviations in selected spectral ranges. In particular, the standard deviation of 52 samples of $\widehat{D}''_S$ (25 cm$^{-1}$-wide bins) is calculated across the spectrum and reported in the summary file for monitoring purposes. Figure 22 depicts this low-
resolution proxy of the noise-equivalent spectral radiance (NESR), also called NEN in the AERI documentation, for a clear-sky view. All those summary data are recorded once per sky view and the associated file size is normally 3 MB per day. Finally, the raw data (including all acquired 16-bit interferograms, $\approx$ 8.2 GB per day) are recorded and compressed by the processing

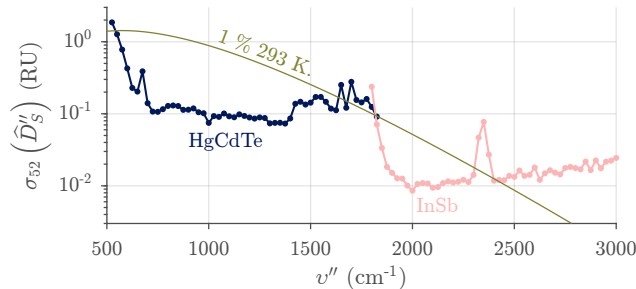

**Figure 22.** Typical "sky NEN" measured with ASSIST-22 and calculated as a moving standard deviation over 52 spectral samples (25 cm$^{-1}$-wide bins) of the imaginary radiance $\widehat{D}''_S$. The illustrated dotted curves are similar to the NESR calculated for a 12.6 s dwell time and 0.48 cm$^{-1}$ bin spacing and depend on fluctuating sky and ABB radiances. The smooth curve represents 1 % of a 293 K Planck curve. RU = mW (m$^2$ sr cm$^{-1}$)$^{-1}$.

software in case reprocessing is later required, but they are automatically deleted, starting from the oldest files and on a daily basis, once the drive associated to the base output directory approaches saturation.

## 4  Validation

A brief comparison between ASSIST-18 and AERI-06 was performed in September 2023 at the NOAA David Skaggs Research Center (DSRC, Boulder, Colorado). During the comparison, the two IRS were separated by approximately 250 m, the AERI was roughly 10 m higher than the ASSIST, and the sky dwell time was 19 s for the AERI and 13 s for the ASSIST. Both instruments were similarly designed to measure spectral radiance with an absolute accuracy better than "1% of the ambient radiance" (Knuteson et al., 2004a). For an ambient radiance represented as a 293 K Planck curve, the mean difference between any pair of ASSIST-AERI should thus be zero within $\pm 1.4$ % ($\pm\sqrt{2}$ %) of the 293 K Planck curve. Figure 23 shows that this is mostly true for ASSIST-18 and AERI-06 in the HgCdTe channel. There are statistically significant spiking differences between the two instruments in the ranges from 525 to 620 cm$^{-1}$, 725 to 805 cm$^{-1}$, and 1110 to 1230 cm$^{-1}$, most likely caused by differences in residual instrument line shapes (the ILS that remains after a necessarily imperfect FFOV ILS correction and resampling) coupling through a rapidly varying spectral radiance in the vicinity of atmospheric absorption lines. This idea is supported by the observation that such spikes are not present when a reference blackbody, of the same design as the ASSIST's ABB and HBB and emitting a smooth and featureless radiance, is installed in the zenith port of each instrument in sequence. The spikes in the 1400 to 1800 cm$^{-1}$ region are visible for all types of scenes and are instead caused by strong internal water vapor absorption (Rowe et al., 2011a) in both instruments, though in this specific case the effect is worse for AERI-06.

Figure 24 illustrates that the differences observed in the HgCdTe channel have minimal impact on the thermodynamic profiles retrieved by TROPoe from a subset of the channel's spectral bins: the temperature profiles and the water vapor mixing ratio profiles agree well within the combined uncertainties at all relevant heights above ground level. Figure 25 finally shows how the two IRS compare in the InSb channel. The conclusions are similar as for the HgCdTe channel: the measurements

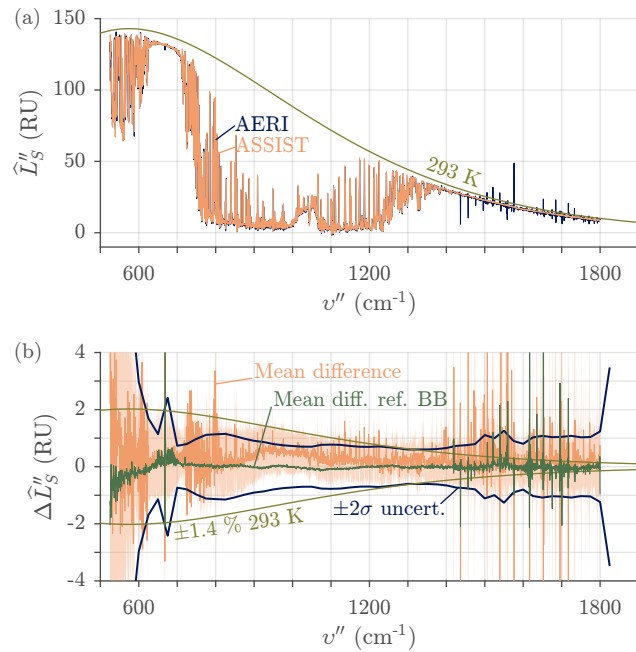

**Figure 23.** Comparison between ASSIST and AERI measurements in the HgCdTe channel (ASSIST-18 and AERI-06 co-located at DSRC, Colorado, on September 23 2023). (a) Clear sky radiance, single view at 0800 UTC. The smooth curve is a reference 293 K Planck curve. (b) Statistics of the radiance differences over the same day (2331 valid clear-sky samples): the solid orange line is the mean difference $\mu$, the shaded region represents $\mu \pm 2\sigma$ from all radiance difference samples. The solid dark blue line is the $\pm 2\sigma$ total uncertainty computed as the root sum square of the individual ASSIST's and AERI's "sky NEN" estimates (see Fig. 22); in this region, the AERI uncertainty is 2 to 3 times higher than the ASSIST uncertainty and therefore dominates the total uncertainty. The smooth curve represents $\pm 1.4$ % of the 293 K Planck curve. Finally, the dark green line labeled as "Mean diff. ref. BB" is the mean difference measured when a reference blackbody at $44.65^\circ$C is sequentially installed in the zenith port of each instrument (distinct validation experiment). RU = mW (m$^2$ sr cm$^{-1}$)$^{-1}$.

are in general agreement, but there are statistically significant differences where the derivative of the radiance with respect to
wavenumber is high, which suggests differences in residual ILS. A more detailed diagnosis establishing how the AERI and
the ASSIST each contribute to the observed difference in the absolute would require a sequential comparison with an adequate
spectral standard, which was not done during this short-duration experiment.

A longer validation experiment with ASSIST-22 was performed from August to November 2023 at the upper-air station in
Stony Plain, Alberta (71119 WSE). Although this experiment will be the subject of a future publication once data analysis
is completed, Fig. 26 presents an example of how the clear-sky radiance measured by the ASSIST compares to the optimal
radiance estimate provided by TROPoe (forward calculation). Internally, TROPoe uses the Line-By-Line Radiative Transfer
Model (LBLRTM) (Clough et al., 2005) as a forward model with degradation by a truncation ILS which is identical for AERI
and ASSIST. Despite its lower vertical resolution, the thermodynamic profile retrieved by TROPoe is in good agreement

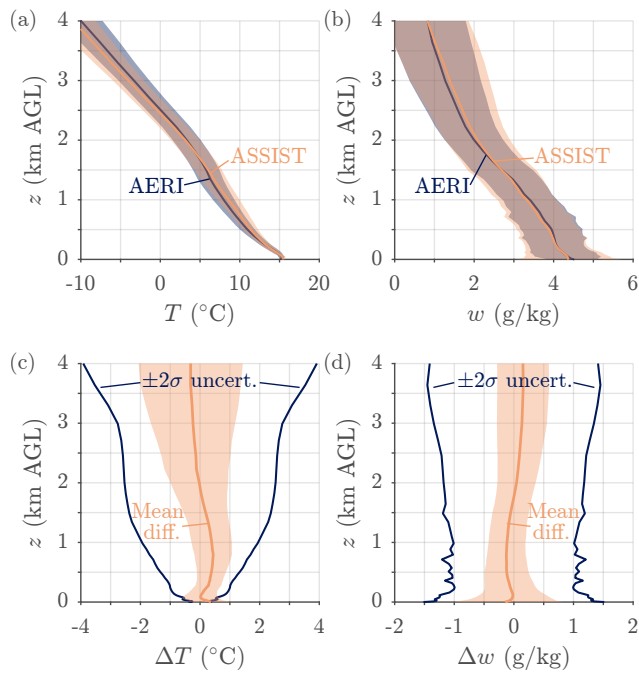

**Figure 24.** Comparison of thermodynamic profiles retrieved by TROPoe from the downwelling radiances measured by ASSIST and AERI and shown in Fig. 23 (one profile every 5 minutes, computed from the closest radiance measurement without coadding). (a) Temperature profiles ($T$) at 0800 UTC with shaded $\pm 2\sigma$ uncertainty provided by TROPoe. (b) Same as (a) for the water vapor mixing ratio ($w$). The retrieved precipitable water vapor and liquid water path were 0.9 cm and 0.2 g m$^{-2}$, respectively. (c) Statistics of the temperature differences over the same day (260 valid samples). The solid line is the mean difference $\mu$, the shaded region represents $\mu \pm 2\sigma$ from all temperature difference samples. The solid dark blue line is the $\pm 2\sigma$ total uncertainty computed as the root sum square of the individual TROPoe uncertainty estimates, which are nearly identical for both instruments. (d) Same as (c) for the water vapor mixing ratio. AGL: above ground level.

with that measured by a radiosonde (RS) launched from the upper-air station at 1200 UTC, with both sensors capturing the
600 temperature inversion, while the associated radiance residuals fall well within the observational uncertainty.

## 5 Conclusions

The ASSIST is a commercially available infrared spectroradiometer designed for automated ground-based downwelling radiance measurements. It essentially consists in a Fourier-transform spectrometer and a radiometric calibration module housed in a field-deployable environmental enclosure. In scripted mode, the ASSIST operates on a repeating view schedule, bracketing
sky views (zenith angle) with periodic calibration views (two integrated blackbodies) and recording the associated sequence of interferograms in two complementary detection channels (HgCdTe and InSb). An integrated software processes batches of interferograms in near real-time to produce absolute radiance spectra that are calibrated and corrected for instrument-specific

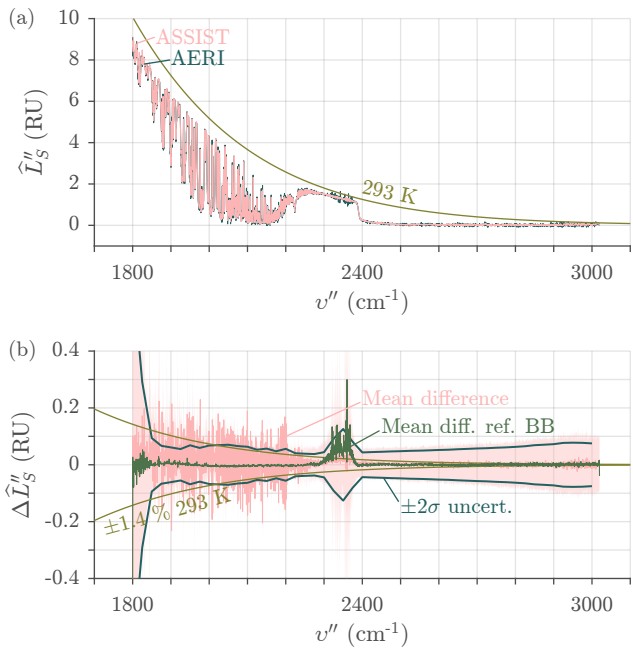

**Figure 25.** Same as Fig. 23, but for the InSb channel instead of the HgCdTe channel. RU = mW $(m^2$ sr $cm^{-1})^{-1}$.

distortions. By default, the primary data output is a time series of discrete and cropped radiance spectra (13 s dwell time per processed spectrum) with a 0.5 cm$^{-1}$ bin spacing in the 525 to 3300 cm$^{-1}$ spectral range.

The ASSIST is built for continuous and unattended operation over long periods and in remote locations: in addition to its automated acquisition and processing, it automatically protects itself from precipitation and dust, its infrared detector does not require cooling with cryogens, it has an auto-restart capability, it automatically deletes the oldest raw data as it approaches disk saturation, it can maintain its radiometric performance in a variety of climates and weather conditions, its modular design facilitates maintenance and reduces the need for interventions, and it constantly logs data useful to monitor its status, track

changes in the immediate environment, and diagnose potential problems remotely. For these reasons and others, a large and growing number of quality observations have been collected by field-deployed ASSIST instruments.

    At the time of this writing, several improvements are being considered based on feedback from end-users and analyses of maintenance and service cases. On the hardware side, these include a new precipitation sensor based on optical detection and counting of falling hydrometeors, a systematic integration of an uninterruptible power supply (UPS) to the electrical interface,

and better solar radiation shielding of the mast sensor to minimize solar bias. Several improvements to the routine processing are also under evaluation, for instance the correction of in-band low-responsivity bins caused by internal air absorption (Rowe et al., 2011a), better correction of the ILS associated to a chromatic and nonrectangular FOV (Desbiens et al., 2006a), pre-correction of electrical dispersion, and better estimation of the ZPD amplitude for the NLC (Lachance and Rochette, 2000).

    A follow-up paper is under preparation to describe the spectro-radiometric performances of the ASSIST under different

operating conditions, including an intercomparison of several co-located ASSIST units.

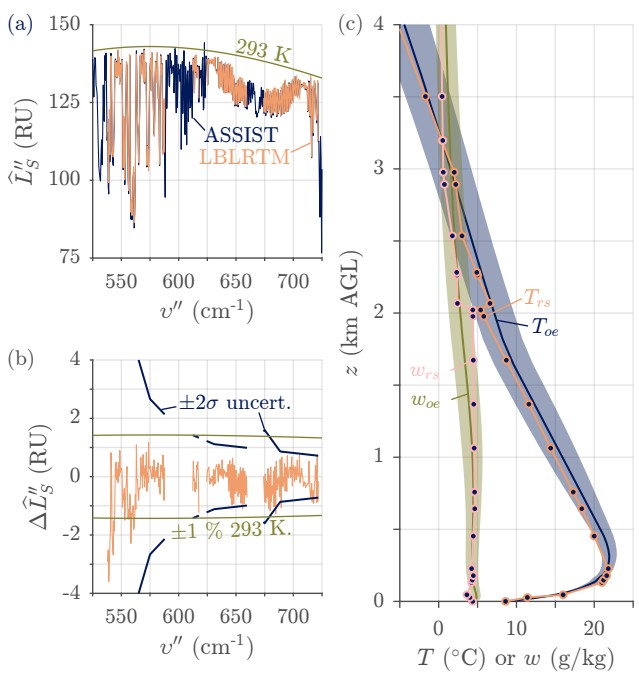

**Figure 26.** Comparison between an ASSIST measurement and LBLRTM (ASSIST-22 on October 9 2023, 1200 UTC, in Stony Plain, Alberta). (a) Spectral radiance for clear-sky conditions, ASSIST measurement versus forward calculation (LBLRTM). The forward calculation is computed from the thermodynamic profile retrieved by TROPoe only in the spectral windows that are selected for high sensitivity to temperature and water vapor (Turner and Blumberg, 2018). The smooth curve is a reference 293 K Planck curve. (b) Residuals between the measurement and the forward calculation. The solid dark blue line is the $\pm 2\sigma$ uncertainty model used by TROPoe, an inflated version of the ASSIST "sky NEN" (Turner and Löhnert, 2014). The smooth curve represents $\pm 1$ % of the 293 K Planck curve. (c) Retrieved optimal estimates of the temperature profile ($T_{oe}$) and water vapor mixing ratio profile ($w_{oe}$) (solid lines with shaded $\pm 2\sigma$ uncertainty) in the first 4 km above ground level compared to the in-situ thermodynamic profile ($T_{rs}$, $w_{rs}$, solid lines with round markers) measured by the RS. The retrieved precipitable water vapor and liquid water path were 1.2 cm and 0.5 g m$^{-2}$, respectively. AGL: above ground level. RU = mW (m$^2$ sr cm$^{-1}$)$^{-1}$.

*Data availability.* The data used to build the figures shown in this paper are available from the corresponding author upon reasonable request.

*Author contributions.* VMB: data curation, formal analysis, visualization, and original draft preparation. JL: preparation of Fig. 3. DDT: TROPoe analyses. All authors contributed to the design of the spectroradiometer and/or its signal processing software and helped prepare and review the manuscript.

*Competing interests.* Most authors are employed by or consult for the company that manufactures the instrument described in this paper.

*Disclaimer.* The results and conclusions, as well as any views or opinions expressed herein, are those of the authors and do not necessarily reflect those of NOAA or the U.S. Department of Commerce.

*Acknowledgements.* The authors acknowledge the effort of many people at LR Tech for the continuous development and improvement of the ASSIST and the preparation of this manuscript: Jérôme Asselin, Josée Boisvert, William Bouchard, Charles Brillon, Jérôme Chabot, Davit Danielyan, Alexandre Girardin, André Lanouette, Carlos Andres Ospina, Éric Simard Turgeon, and others. The authors also thank the NOAA-PSL and NREL-WES teams for their questions and suggestions in relation to the ASSIST and Jonathan Gero from the University of Wisconsin - Madison for providing the AERI data.

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
