# Peer review of "The Atmospheric Sounder Spectrometer by Infrared Spectral Technology (ASSIST): Instrument design and signal processing"

_EGUsphere, 2024_

## Referee Comment (RC1)

This is a complex paper to read. the authors' effort at presenting a complete view of the ASSIST theory and development is greatly appreciated. In my opinion, the paper would be improved by focusing on those details they consider most important in section 3, but then they are best equipped to make that evaluation. Notwithstanding the statement that a follow-up paper in preparation that describes the performance of ASSIST, this paper would greatly benefit from the presentation of actual ASSIST spectral radiance spectra compared to co-located AERI measurements and also to well known forward models such as LBLRTM. That is still the best way for the reader to evaluate whether or not the considerable effort on the part of the ASSIST team has produced a capable system that accurately measures downwelling radiance. Presenting a pair of cases, one of a warm, wet atmosphere and the second, a cold, dry atmosphere, would nicely span the spectral and environmental performance regime.

**Comments/Corrections**

Pg 2   line 28 why refer to the signatures of liquid water, ice and aerosols as "spectrally dependent"? Do you perhaps mean that they are broad and don't offer much spectral structure?

Pg 2, lines 40-45 references imply use alongside AERI instruments, however, none of the references checked provide a comparison between the two

pg 3. Section 2.1   Is the ASSIST (the interferometer) an FTS built by LR Tech? It would be a good idea to state so, or otherwise state the origin of the FTS.

Pg 5. line 95 "… move together, in opposite directions relative to the beamsplitter." What are the "directions relative to the beamsplitter"? Does opposite refer to both moving away from the beamsplitter (as traveling east is opposite from traveling west), or one moving "in" towards the beamsplitter while the other moves "out" away from the beamsplitter?

Pg 5. line 103. So there are two photodetectors, one for each polarization. Is there also a dedicated preamplifier for each photodetector?

Pg 5. line 110 scan velocity of *approximately* 2.0 cm s$^{-1}$? Can you specify the physical motion that corresponds to and OPD span of +/- 1.04 cm?

Pg 5. line 115  Are you saying that essentially the same metrology fringe is typically selected scan after scan? In other words, there is typically no, or minimal, "jitter" in the fringes? How do you determine this?

Pg 6.  Figure 3 caption. I would guess that this is actually a filter placed at an angle that reflects wavenumbers below a cutoff and transmits those above? Same comments apply to section 2.3. More detail about the detector and cooler setup would be interesting.

Pg 8. Table 2, spell out entr. Entrance pupil area, what does this refer to? Diameter of the FS? The entrance pupil as described as near the CC apices? Data rate would make more sense as the number of seconds per interferogram. Could easily be stated as both.

Pg 10. Section 2.5. Quite specific information is provided about several of the components in the calibration system, but nothing is specified about the temperature controller. For example, at what rate does the controller monitor the BB readings? There is a 2 minute time constant quote for closed loop

adjustments, how often does the controller fine-tune that adjustment?  If ambient conditions change, and the HBB needs to adjust, how is that process monitored and reported?

Pg 11. Section 2.6 Environmental Enclosure.  According to various specifications presented, you have an operating range from -25C to 40C, described as "harsh".  This is a subjective evaluation, and these values are rather milder than several use cases in which other instruments are performing adequately.

Pg 14.  Line 246.  When you say "a low impedance path between chassis and ground" are you referring to the grounding connection for your power supply such that you enact a single point ground for the system?

Pg 15.  Figure 9. I would prefer to see a legend on the plot describing the various traces, rather than having to read the caption.  If available, a completely independent air temperature reading would also be of interest (ie.  not recorded by the ASSIST). Would be useful in particular for understanding the ABB temperatures

Pg 16.  Table 4  I would dispense with the use of (-) to denote a dimensionless quantity, or a quantity not having intrinsic units.  At least that is what it seems to be for.  For the cooler Power up count - what is that?  If you track hours of operation that would be useful information.  What is the value of the quantity $T_A/T_H$?  Are you perhaps reporting one of the mean $T_A$ and the mean $T_H$?  If so, replace the '/' with or.  For the hatch status, do you have actual limit switches whose state is read to determine open, closed, or other?

Pg 16, line 270.  To clarify, the FOV corrections, resampling and cropping are carried out AFTER averaging forward and reverse scans? Those would seem to be characteristics that could potentially be different in the forward or reverse direction.  This could really use more detail.

Pg 19. line 336, do you perhaps need to move the  commas?  For example:

Fitted values of the factor $2a_2V_{0,Hd}$, for HBB views normally associated to the highest photon irradiance for a dozen ASSIST instruments, are distributed around 0.10 with extreme values reaching 0.04 and 0.16.

pg 19. line 238 remove the ( ) around the reverse scan description and write it as a stand-alone sentence.

Pg 25/26  and figure 16. As no two days are the same, and no two sites the same, is there any particular relevance to showing these signals evolving over a day? Showing cases near the limits might provide more information.

Pg 26. line 427, something doesn't make sense in the wording

pg 28. figure 17.  I am unsure how to interpret the value of the "first moment".  Is that the median of the area under the curve?  Center of mass means what in this context?

Pg 29. Line 464  "… the relative spectral shift is spectrally dependent" … does this mean that the spectral shift varies with wavelength?  Some other characteristic of the spectrum?

Pg 29. Line 472.  Perhaps you could comment briefly on the most critical facet of the detector alignment as determined by your new procedure?

Pg 30. Figure 18.  Is the radiance represented here actual measurements from the ASSIST-22, or modeled radiance based on the characteristics of ASSIST-22?

Pg 30.  Line 492 "typically displays strong out-of-band noise"  how is this happening?  Are there not optical filters, electronic filters?  Or what is being referred to here as out-of-band noise?

Pg 34.  Figure 22.  Looks like a blue and red curve on my screen.  You might want to refer to line type or some other line or plot characteristic to avoid color confusion.

---

## Author Response (AR1)

**Response to reviewers' comments**

We thank the two anonymous reviewers for their comments and their suggestions, which helped improve the manuscript. In the following we provide a point-to-point response to all reviewer comments. The reviewers' comments are printed in italic and our response in roman font type. The changes we made to the manuscript are highlighted in red. For the reviewers' convenience we also copied larger changes we made to the manuscript to this response and enclosed them with quotation marks.

**REVIEWER 1**

*This paper describes the ASSIST instrument, a Fourier transform spectrometer operating in the infrared and designed for ground-based measurements of downwelling radiances. The paper is clearly written, and the instrument as well as the data processing are well described. I have a few questions/comments, though, which are listed below.*

**Response:** Thank you for this positive evaluation.

**Comments/Corrections**

1. *(page 3, line 83) You describe the beamsplitter as "nominally flat". Don't you have problems with multiple reflections at parallel surfaces?*

   **Response:** There are indeed multiple reflections at the parallel surfaces of the beamsplitter. They cause spectral modulation or "channeling" (imagine a low-finesse Fabry-Perot glass etalon versus a glass slab with poor parallelism). But since those reflections are nominally the same for both arms of the interferometer (cancelled dispersion), since the beamsplitter is optically thick relative to the maximum OPD, and critically since the beamsplitter's properties are identical for calibration views and sky views, there are no problematic outcomes after radiometric calibration. Please see this conference paper for a brief description and relevant references: https://doi.org/10.1364/FTS.2001.FWB4.

2. *(page 6, Fig. 3) "For simplicity, rays are traced at 550 and 1650 cm$^{-1}$ for only one arm of the interferometer and from two field points located at infinity: zero and 21 mrad relative to the optical axis (blue and red, respectively)". With this description I'd expect four beams: 550 cm$^{-1}$ with zero and 21 mrad and 1650 cm$^{-1}$ with zero and 21 mrad. How do I know which rays belong to 550 cm$^{-1}$ and which rays belong to 1650 cm$^{-1}$?*

   **Response:** The color-coding is only used to distinguish the two propagation angles (0 and 21 mrad). Each beam of rays shown in the figure should be considered to be bichromatic (550 and 1650 cm$^{-1}$ components overlapped) until separation by the dichroic beamsplitter in the dewar. After separation, the four beams are visible in the dewar (please see the inset). We agree this can be confusing and have made the following modification:

   (page 6, Fig. 3) For simplicity, bichromatic rays, with 550 and 1650 cm$^{-1}$ components overlapping until the DBS, are traced  for only one arm of the interferometer and from two field points located at infinity: zero and 21 mrad relative to the optical axis ( blue and pink, respectively).

3. *(page 10, line 206) "A NIST-traceable in-house calibration procedure". Can you spend a few words on this calibration procedure? What is the reference?*

   **Response:** Yes, we have added the following short description:

(page 10, line 205) The six lookup tables needed by a given calibration module are not swappable and are built by LR Tech through a NIST-traceable in-house calibration procedure during which the assembled thermistors are immersed in a circulating glycol bath whose temperature is set to span the -27 to 63°C range with reference to a standard platinum resistance thermometer. ;they The tables are saved…

4. *(page 11, line 215) "Section 33.3" must be "Section 3.3".*
   **Response:** That's right. The corrections have been made (there were several instances of a similar error).

5. *(page 11, line 221) "...a modular air conditioning unit that can maintain a steady-state internal temperature for external temperatures up to at least 40°C". Which steady-state internal temperature is used?*
   **Response:** The standard air conditioning "cooling setpoint" is 32°C. We have added this information:

   (page 11, line 221) …a modular air conditioning unit that can maintain a steady-state internal temperature close to 32°C for external temperatures up to at least 40°C.

6. *(page 18, line 304) Do you have any idea if the non-linearity may change over the lifetime of the instrument due to detector aging? Could the modulation efficiency change over time? Are laboratory characterization measurements for the non-linearity coefficients foreseen from time to time in order to monitor and to account for such possible changes?*
   **Response:** We do not suggest a detailed characterization at the factory on a regular basis, but we do suggest a validation with a reference blackbody on an annual basis. This validation, which can be performed in the field, is sufficient to detect if an ASSIST is drifting outside the acceptable relative accuracy range for any reason, including a drift in nonlinearity parameters. In our experience, drifts in nonlinearity parameters are mainly associated to the drop in raw signal level caused by a dirty scene mirror (the radiometric calibration itself is still accurate since the scene mirror is inside the calibration path, but the nonlinearity correction is affected since it depends on the absolute signal level). This can be partially cancelled by cleaning the scene mirror, or completely cancelled by replacing the scene mirror. We have not seen evidence of detector aging or progressive changes in modulation efficiency over the past decade, though we have also not specifically set out to detect such evolution. We appreciate the reviewer's question and the opportunity to provide details in this response, but prefer not to include such details in the manuscript.

7. *(page 22, line 370) "Figure 13 shows the typical responsivity (magnitude of the complex gain) in the 500 to 3000 cm$^{-1}$ range for the forward scan direction". Just out of curiosity: I understand that the phase of the gain is sweep direction dependent, but is the magnitude the same for both sweep directions?*
   **Response:** Yes, the magnitude is the same for both sweep directions, at least within the measurement noise. We have stated this explicitly:

   (page 22, line 370) Figure 13 shows the typical responsivity (magnitude of the complex gain) in the 500 to 3000 cm$^{-1}$ range for the forward scan direction; the responsivity for the reverse scan direction is the same within the measurement noise.

8. *(page 23, Fig. 13) How do you treat the absorption lines in the response curves? I assume that these absorption lines lead to erroneous radiances in the calibrated data? The same question holds for the atmospheric lines in the instrument offset (Fig. 14).*

   **Response:** No special signal processing is currently performed to avoid this phenomenon. For weak internal absorption lines (assuming stable properties for the duration of a calibration cycle), the spectral bins under the lines are unbiased but show a degraded uncertainty with respect to neighboring bins (higher random noise after radiometric calibration). For strong internal absorption lines, the affected bins can become biased with extreme uncertainty. This is somewhat explained in the two sentences starting at line 404. The hardware solution is to minimize the amount of absorbing gas in the calibration path (e.g. with a desiccant, as is done for the ASSIST, or by continuously purging the enclosure with dry nitrogen). However, a significant part of the calibration path is open to the atmosphere in the ASSIST, so this solution is limited in its application. A software solution is presented in https://doi.org/10.1364/OE.19.005930, which is cited in the manuscript, but it is not currently implemented for the ASSIST. Please see the comment about this in the conclusion, line 566. Finally, the response to Reviewer 3's comment #24 should also make this point clearer.

9. *(page 26, Fig. 16(a)) What are the red dots in the curve?*

   **Response:** Thank you for pointing out this omission. The editorial support team asked us to revise the colour scheme used in the maps and charts to allow readers with colour vision deficiencies to correctly interpret the findings, which we did right after submission (the Reviewer had the unrevised submitted manuscript in hand). We updated all captions accordingly and also used this opportunity to add labels to some plots to minimize reliance on captions. The up-to-date version of Fig. 16 and its caption are shown below. We have removed the red dots which represented the temperature readout from a nearby meteorological station (as in Fig. 9).

[Figure]

"**Figure 16.** (a) Typical temperatures involved in radiometric calibration over a full day of uninterrupted outdoor operation (ASSIST-22, June 16 2024): HBB temperature ($T_H$), ABB temperature ($T_A$, the peak temperature variation is 165 mK over 132 s), reference port temperature ($T_R$), and reflected temperature ($T_r$). Also shown are the external temperature ($T_{ext}$) and the brightness temperature in the opaque 675-680 cm$^{-1}$ region ($T_{B,air}$), representative of the temperature of the air in the first few meters above the instrument. (b) Sampled responsivity associated to a forward scan near $\bar{v}$ = 1000 cm$^{-1}$ (HgCdTe channel) and near $\bar{v}$ = 2500 cm$^{-1}$ (InSb channel) for the same period. The hatch closed automatically due to precipitation for a brief period around 1840 UTC (shaded gray). (c) Transition from hatch cover's inner surface radiance (1843 UTC) to sky radiance after the precipitation event (1844 UTC) in the HgCdTe channel. The smooth curve is a fitted Planck curve representative of the hatch cover's inner surface temperature. RU = mW (m$^2$ sr cm$^{-1}$)$^{-1}$."

10. *(page 26, line 428) "...is proportional the cosine" should read "...is proportional to the cosine".*
    **Response:** That's right. The correction has been made.

11. *(page 33, Fig. 21) The red line turns into yellow around 1750 cm-1 and becomes white between 1825 and 1900 cm-1 on my screen. This is a bit irritating.*

**Response:** Agreed. The white line was voluntarily added to show how the average values match in the overlap region, but it was never described in the caption. The up-to-date version of Fig. 21 and its caption are shown below. The confusing white line has been entirely removed.

[Figure]

"**Figure 21.** (a) Typical final radiance in the InSb channel and for a clear sky view, measured with ASSIST-22 (high FFOV-distortion). The curves below 1825 cm$^{-1}$ are the tails of the measurements in the HgCdTe channel, with and without NLC, smoothed in both cases to better expose the overlap with the InSb signal. The smooth curve is a 293 K Planck curve. (b) Associated finite field of view correction. The smooth curves represent $\pm 1$ % of the 293 K Planck curve. RU = mW (m$^2$ sr cm$^{-1}$)$^{-1}$."

**REVIEWER 3**

*This is a complex paper to read. The authors' effort at presenting a complete view of the ASSIST theory and development is greatly appreciated. In my opinion, the paper would be improved by focusing on those details they consider most important in section 3, but then they are best equipped to make that evaluation. Notwithstanding the statement that a follow-up paper in preparation that describes the performance of the ASSIST, this paper would greatly benefit from the presentation of actual ASSIST spectral radiance spectra compared to co-located AERI measurements and also to well-known forward models such as LBLRTM. That is still the best way for the reader to evaluate whether or not the considerable effort on the part of the ASSIST team has produced a capable system that accurately measures downwelling radiance. Presenting a pair of cases, one of a warm, wet atmosphere and the second, a cold, dry atmosphere, would nicely span the spectral and environmental performance regime.*

**Response:** Thank you for this evaluation. Although this is outside the scope we initially had in mind for this manuscript, we agree that a comparison with known references or standards would be beneficial. It is fortunate that we already have some data in hand to perform such a comparison. Therefore, we have added four new figures. The first three figures, shown below, are concerned with the AERI-ASSIST comparison (ASSIST-18, not the usual ASSIST-22): HgCdTe channel (Fig. 23), InSb channel (Fig. 25),

and thermodynamic profiles generated from the HgCdTe channel data using TROPoe (Fig. 24). In all cases, we show snapshot measurements performed at 0800 UTC on 2023-09-23 in Boulder, Colorado. Moreover, we show difference statistics (mean and standard deviation for valid radiance difference samples) for the full duration of the same day. The sky above the instruments was free of clouds for most of the day and the atmosphere was relatively cold and dry (surface temperature between 13 and 23°C, surface water vapor mixing ratio between 2 and 7 g/kg). Given the limited presentation space, these are challenging comparison conditions we consider to be interesting since they entail a large in-spectrum dynamic range. In any case, the two instruments were co-located in Boulder only for a few days in September 2023 and we do not have the data required to present the case of a warm and wet atmosphere as suggested by the reviewer.

[Figure]

"**Figure 23.** Comparison between ASSIST and AERI measurements in the HgCdTe channel (ASSIST-18 and AERI-06 co-located at DSRC, Colorado, on September 23 2023). (a) Clear sky radiance, single view at 0800 UTC. The smooth curve is a reference 293 K Planck curve. (b) Statistics of the radiance differences over the same day (2351 valid clear-sky samples): the solid line is the mean difference $\mu$, the shaded region represents $\mu \pm 2\sigma$ from all radiance difference samples. The solid dark blue line is the $\pm 2\sigma$ total uncertainty computed as the root sum square of the individual ASSIST's and AERI's "sky NEN" estimates (see Fig. 22); in this region, the AERI uncertainty is 2 to 3 times higher than the ASSIST uncertainty and therefore dominates the total uncertainty. The smooth curve represents $\pm 1.4$ % of the 293 K Planck curve. Finally, the dark green line labeled as ``Mean diff. ref. BB" is the mean difference measured when a reference blackbody at 44.65°C is sequentially installed in the zenith port of each instrument (distinct validation experiment). RU = mW (m$^2$ sr cm$^{-1}$)$^{-1}$."

[Figure]

"**Figure 24.** Comparison of thermodynamic profiles retrieved by TROPoe from the downwelling radiances measured by ASSIST and AERI and shown in Fig. 23 (one profile every 5 minutes, computed from the closest radiance measurement without coadding). (a) Temperature profiles ($T$) at 0800 UTC with shaded $\pm 2\sigma$ uncertainty provided by TROPoe. (b) Same as (a) for the water vapor mixing ratio ($w$). The retrieved precipitable water vapor and liquid water path were 0.9 cm and 0.2 g m$^{-2}$, respectively. (c) Statistics of the temperature differences over the same day (260 valid samples). The solid line is the mean difference $\mu$, the shaded region represents $\mu \pm 2\sigma$ from all temperature difference samples. The solid dark blue line is the $\pm 2\sigma$ total uncertainty computed as the root sum square of the individual TROPoe uncertainty estimates, which are nearly identical for both instruments. (d) Same as (c) for the water vapor mixing ratio. AGL: above ground level."

[Figure]

"**Figure 25.** Same as Fig. 23, but for the InSb channel instead of the HgCdTe channel."

The fourth figure, Fig. 26 shown below, presents a comparison between an ASSIST measurement and the Line-By-Line Radiative Transfer Model (LBLRTM). The thermodynamic profile that feeds LBLRTM is the profile retrieved by TROPoe from the ASSIST measurement. LBLRTM is only evaluated in those spectral bins that are used by TROPoe. In panel (c), the retrieved profile is compared with the profile measured by a co-located radiosonde. The conditions are once again relatively cold and dry, but this time there is a temperature inversion.

[Figure]

"**Figure 26.** Comparison between an ASSIST measurement and LBLRTM (ASSIST-22 on October 9 2023, 1200 UTC, in Stony Plain, Alberta). (a) Spectral radiance for clear-sky conditions, ASSIST measurement versus forward calculation (LBLRTM). The forward model is computed from the thermodynamic profile retrieved by TROPoe only in the spectral windows that are selected for high sensitivity to temperature and water vapor (Turner and Blumberg, 2018). The smooth curve is a reference 293 K Planck curve. (b) Residuals between the measurement and the forward calculation. The solid dark blue line is the $\pm 2\sigma$ uncertainty model used by TROPoe, an inflated version of the ASSIST "sky NEN" (Turner and Löhnert, 2014). The smooth curve represents $\pm 1$ % of the 293 K Planck curve. (c) Retrieved optimal estimates of the temperature profile ($T_{oe}$) and water vapor mixing ratio profile ($w_{oe}$) (solid lines with shaded $\pm 2\sigma$ uncertainty) in the first 4 km above ground level compared to the in-situ thermodynamic profile ($T_{rs}$, $w_{rs}$, solid lines with round markers) measured by the RS. The retrieved precipitable water vapor and liquid water path were 1.2 cm and 0.5 g m$^{-2}$, respectively. AGL: above ground level. RU = mW (m$^2$ sr cm$^{-1}$)$^{-1}$. "

We have added a new "Validation" section to the manuscript to introduce the new validation figures:

[revised manuscript text omitted]

This publication is now part of the bibliography:

Clough, Shepard A., et al. "Atmospheric radiative transfer modeling: A summary of the AER codes." *Journal of Quantitative Spectroscopy and Radiative Transfer* 91.2 (2005): 233-244.

We have also added a sentence to the acknowledgments:

(page 35, line 580) The authors also thank the NOAA-PSL and NREL-WES teams for their questions and suggestions in relation to the ASSIST and Jonathan Gero from the University of Wisconsin – Madison for providing the AERI data.

**Comments/Corrections**

12. *(page 2, line 28) Why refer to the signatures of liquid water, ice and aerosols as "spectrally dependent"? Do you perhaps mean that they are broad and don't offer much spectral structure?*
    **Response:** No, we simply meant to say that these show the necessary variation of radiative properties versus wavenumber. We recognize the formulation is confusing. We have changed to "spectral signature", which is the conventional way to describe the same concept:

    (page 2, line 27) One of the strengths of the thermal infrared is that it contains the absorption bands of important atmospheric gases in addition to the  spectral signatures of liquid water, ice, and aerosols.

13. *(page 2, lines 40-45) References imply use alongside AERI instruments, however, none of the references checked provide a comparison between the two.*
    **Response:** We agree the text seems to imply that the associated references provide a comparison between the two IRS, which is misleading. We have removed the comment about the use alongside AERI:

    (page 2, line 42) In recent years, this standard version of the ASSIST has been deployed to provide thermodynamic profiles of the atmospheric boundary layer

14. *(page 3, Sec. 2.1) Is the ASSIST (the interferometer) an FTS built by LR Tech? It would be a good idea to state so, or otherwise state the origin of the FTS.*
    **Response:** The whole ASSIST instrument, including its interferometer, is designed and built by LR Tech (obviously, LR Tech integrates parts from a plethora of suppliers). This is already stated at line 2, but it is worth insisting as the reviewer suggests:

    (page 2, line 37) This paper introduces a new ground-based IRS entirely built by LR Tech, the Atmospheric Sounder by Infrared Spectral Technology (ASSIST), which follows design, measurement, and processing philosophies…

15. *(page 5, line 95) "… move together, in opposite directions relative to the beamsplitter." What are the "directions relative to the beamsplitter"? Does opposite refer to both moving away from the beamsplitter (as traveling east is opposite from traveling west), or one moving "in" towards the beamsplitter while the other moves "out" away from the beamsplitter?*
    **Response:** We agree this sentence is not clear. We have made the following modification to address the issue:

(page 5, line 95) The opto-mechanical gain of this interferometer (the ratio between the OPD and the physical displacement of a retroreflector) is 4 since both retroreflectors  are displaced when the swing arm rotates, with one retroreflector moving towards the beamsplitter when the other retroreflector moves away from it .

16. *(page 5, line 103) So there are two photodetectors, one for each polarization. Is there also a dedicated preamplifier for each photodetector?*

    **Response:** Yes. This was implied with the plural "preamplifier circuits", but the reviewer is right that this is worth stating clearly:

    (page 5, line 103) …the associated fringes are measured by a laser detector (not illustrated: one photodiode and one preamplifier circuit for each laser polarization ), providing in-phase and quadrature signals…

17. *(page 5, line 110) Scan velocity of approximately 2.0 cm s$^{-1}$? Can you specify the physical motion that corresponds to and OPD span of +/- 1.04 cm?*

    **Response:** The opto-mechanical gain of the interferometer is stated at line 95 (now with formal definition). We prefer to avoid mentions of mechanical movement to minimize the potential for confusion, but we have still added one example for illustration purposes:

    (page 5, line 110) The total number of interferogram samples at full nominal resolution is 32 768 and the associated OPD span is $\pm1.04$ cm (each cube corner is physically displaced over $\pm\, 0.26$ cm).

18. *(page 5, line 115) Are you saying that essentially the same metrology fringe is typically selected scan after scan? In other words, there is typically no, or minimal, "jitter" in the fringes? How do you determine this?*

    **Response:** The reviewer's interpretation is correct: the same metrology fringe is selected scan after scan. This can be determined by comparing different digital interferograms, either within a given calibration cycle (short term) or between calibration cycles (long term). Visual inspection of the interferogram center burst and/or the spectral phase is sufficient to identify a potential shift by an integer number of fringes (the total number of samples is always 32 768), but the fact that the radiometric calibration yields a zero-mean imaginary radiance also constitutes sufficient proof. We have clarified this point:

    (page 5, line 114) Under normal closed-loop operation of the scanning interferometer, the sampling grid is repeatable from one swing arm scan to the next as there are no fringe count errors (Kleinert et al., 2014) . Direct coadding of sampled interferograms, of the same scan direction and for a stationary input radiation, can therefore be performed to increase the signal-to-noise ratio.

    This publication is now part of the bibliography:
    Kleinert, A., et al. "Level 0 to 1 processing of the imaging Fourier transform spectrometer GLORIA: generation of radiometrically and spectrally calibrated spectra." *Atmospheric Measurement Techniques* 7.12 (2014): 4167-4184.

19. *(page 6, Fig. 3 caption) I would guess that this is actually a filter placed at an angle that reflects wavenumbers below a cutoff and transmits those above? Same comments apply to section 2.3. More detail about the detector and cooler setup would be interesting.*

    **Response:** There are no missing elements in Fig. 3. The reviewer seems to refer to the dichroic beamsplitter (DBS) shown in the inset, which is indeed tilted at 45°. This was not explicitly stated, so we have modified Sect. 2.3 as such:

    (page 7, line 139) Instead of the "sandwich" configuration of the AERI (Knuteson et al., 2004a), the ASSIST detector is based on a "dichroic" configuration: in a vacuum dewar, a dichroic beamsplitter (DBS) with approximate cutoff at 1900 cm$^{-1}$ is tilted at 45° relative to the optical axis and steers the longwave beam (transmitted) and the midwave beam (reflected) to the HgCdTe and InSb chips mounted perpendicular to one another (Fig. 3).

    We consider that the level of detail in Sect. 2.3 is adequate given the scope of the manuscript. Additional details would likely enter the realm of proprietary information without helping the intended audience. We therefore respectfully decline the reviewer's last suggestion.

20. *(page 8, Table 2) Spell out entr. Entrance pupil area, what does this refer to? Diameter of the FS? The entrance pupil as described as near the CC apices? Data rate would make more sense as the number of seconds per interferogram. Could easily be stated as both.*

    **Response:** The entrance pupil is described in lines 129-131. We are using the conventional definition of this concept from optical design theory. A (temporal) rate is conventionally understood to be a number per unit of time and we argue that adding the reciprocal quantity would confuse readers and overcrowd the table. Still, we have made the following modification address the reviewer's first point:

    (page8, table 2)     Optical configuration          Pupil-imaging, no  telescope

21. *(page 10, Sec. 2.5) Quite specific information is provided about several of the components in the calibration system, but nothing is specified about the temperature controller. For example, at what rate does the controller monitor the BB readings? There is a 2 minute time constant quote for closed loop adjustments, how often does the controller fine-tune that adjustment? If ambient conditions change, and the HBB needs to adjust, how is that process monitored and reported?*

    **Response:** For closed-loop operation, the controller operates at a rate of 3.125 Hz. The controller implements a standard proportional-integral-derivative (PID) feedback-based control loop. The difference between the setpoint temperature and the measured temperature is therefore "continuously" (at the 3.125 Hz rate, which is rapid relative to the thermal time constants) monitored and corrected by adjusting the voltage fed to the thin-film heaters. We recognize we provide minimal information about the feedback control and have thus brought the following adjustments:

    (page 9, line 191) To facilitate thermal control, foam sheet insulation is applied between the cavities and their protective stainless steel cases. Near room temperature, the open-loop thermal time constant of the mounted blackbodies is approximately 2 hours..

    (page 10, line 194) A dedicated blackbody controller measures the effective temperature of both cavities and feedback-controls the active heaters .

(page 10, line 197) To this end, all temperature-dependent resistances are first measured using an integrated dual-range ratiometric ohmmeter: > 30.9 kΩ (<-1∘C) for the low-temperature range and < 39.1 kΩ (> -6∘C) for the high-temperature range, with low-temperature-coefficient precision resistors of 500 kΩ and 50 kΩ (0.01%), respectively, acting as internal references. Each measured resistance…

(page 11, line 214) Section 33.3 provides additional information about the cavity emissivity model and the general radiometric calibration. Onboard temperature regulation is achieved using a software-implemented proportional-integral-derivative (PID) controller operating at 3.125 Hz. Under closed-loop operation, the controller continuously adjusts the voltage (0 to 24 V) fed to the BB resistive heaters to minimize the temperature error, which is defined as the temperature setpoint minus the weighted thermistor readout. This is done independently for the two blackbodies. Though the loop parameters can be adjusted by the user through serial communication with the controller, the PID settings are tuned during instrument production so that a small setpoint step leads to a step response with small overshoot, no oscillation, and a closed-loop time constant of approximately 2 minutes. Table 3 summarizes the properties of the blackbody cavities.

22. *(page 11, Sec. 2.6 Environmental enclosure) According to various specifications presented, you have an operating range from -25C to 40C, described as "harsh". This is a subjective evaluation, and these values are rather milder than several use cases in which other instruments are performing adequately.*

**Response:** The reviewer is right to point out that "harsh" is a subjective evaluation that is out of place. We have adjusted the text for a more neutral description:

(page 1, line 5) For atmospheric studies, the ASSIST IRS is integrated into a mobile enclosure enabling autonomous, and reliable ground-based operation under harsh conditions across a range of environmental conditions.

(page 11, line 219) The spectrometer and its calibration module are housed in a urethane-painted aluminum enclosure enabling autonomous field operation under harsh conditions diverse field conditions (Fig. 7).

23. *(page 14, line 246) When you say "a low impedance path between chassis and ground" are you referring to the grounding connection for your power supply such that you enact a single point ground for the system?*

**Response:** No, in this case we meant the physical ground underneath the instrument in the context of lightning protection, not the conceptual electrical ground. We recognize this is not clear and have modified the text as follows:

(page 12, line 245) …integrated stabilizing jacks and spirit level; for field deployment, a grounding electrode accessory can be installed to provides a low-impedance conductive path between the enclosure chassis and the physical ground for lightning protection.

24. *(page 15, Fig. 9) I would prefer to see a legend on the plot describing the various traces, rather than having to read the caption. If available, a completely independent air temperature reading would also be of interest (i.e. not recorded by the ASSIST). Would be useful in particular for understanding the ABB temperatures.*

**Response:** The editorial support team asked us to revise the colour scheme used in the maps and charts to allow readers with colour vision deficiencies to correctly interpret the findings, which we did right after submission (the Reviewer had the unrevised submitted manuscript in hand). We updated all captions accordingly and also used this opportunity to add labels to some plots to minimize reliance on captions. The up-to-date version of Fig.9 and its caption are shown below.

[Figure]

"**Figure 9.** Typical calibration module temperatures over a full day of uninterrupted outdoor operation (ASSIST-22, June 14 2024). (a) External temperature reported by mast sensor ($T_{ext}$), air temperature reported hourly by a nearby  ECCC-MSC meteorological station ($T_{met}$), ABB temperature ($T_A$), and scene mirror ambient temperature ($T_{mir}$). Also shown is the brightness temperature in the opaque 675-680 cm$^{-1}$ region ($T_{B,air}$), derived from the spectrometer sky data and representative of the temperature of the air in the first few meters above the instrument (see Sect. 3.6). The grayed-out area indicates a closed hatch. The meteorological station providing $T_{met}$ is located on the roof of a four-story building approximately 6 km away from the ASSIST, on the other side of the Saint Lawrence River, and its elevation is approximately 35 m higher than that of the ASSIST. (b) HBB temperature readouts relative to the 60°C setpoint. The HBB typically displays a clear vertical temperature gradient. (c) ABB temperature readouts relative to the weighted mean ABB temperature. The top of the ABB cavity was exposed to sunlight between approximately 1600 and 1900 UTC (coordinated universal time), creating a stronger temperature gradient."

In our opinion, the new caption now defining subplot-(a)'s variables between parentheses, e.g. "External temperature reported by mast sensor ($T_{ext}$), …", does ease the plot interpretation and addresses the reviewer's point to some extent. We believe that variable definitions through a subplot legend, in this specific case where several definitions are technical, would overcrowd the figure and impede on visual analysis.

About the *completely independent air temperature reading*, this is already covered: $T_{met}$ is an air temperature reading provided by ECCC-MSC for the Université Laval station, relatively close to the site where the ASSIST was deployed in Lévis, QC. Unfortunately, a co-located external temperature sensor was not available for this specific day of deployment, but in our experience the Université Laval station reading is close (within 1°C) of the reading of a local ground-based temperature sensor that is well vented and shielded from the sun. It is therefore sufficiently representative of the local conditions. In this figure, it offers a reading that matches the brightness temperature measured by the ASSIST in an opaque atmospheric window for several hours.

25. *(page 16, Table 4) I would dispense with the use of (-) to denote a dimensionless quantity, or a quantity not having intrinsic units. At least that is what it seems to be for. For the cooler Power up count – what is that? If you track hours of operation that would be useful information. What is the value of the quantity $T_A/T_H$? Are you perhaps reporting one of the mean $T_A$ and the mean $T_H$? If so, replace the '/' with or. For the hatch status, do you have actual limit switches whose state is read to determine open, closed, or other?*

    **Response:** Agreed. The "(-)" have been removed. The "$T_A/T_H$" has been replaced by "$T_A$ or $T_H$".

    The "cooler power up count" simply states how many times the cryogenic cooler was powered on during its lifetime. Its number of operating hours is not tracked, unfortunately, but we agree this would be useful information.

    For the hatch status, yes there are actual limit switches.

    We have brought the following modifications to the text:

    (page 16, table 4) Lifetime power up count

    (page 14, line 236) The hatch status, inferred from the readings of two limit switches, is also reported in the output files for filtering purposes.

26. *(page 16, line 270) To clarify, the FOV corrections, resampling and cropping are carried out AFTER averaging forward and reverse scans? Those would seem to be characteristics that could potentially be different in the forward or reverse direction. This could really use more detail.*

    **Response:** Yes, the FFOV correction, the resampling, and the cropping are carried out on the real radiance vector $\hat{L}_S$ obtained after forward-reverse averaging (i.e. $\hat{L}_S = \left(\hat{L}_{Sf} + \hat{L}_{Sr}\right)/2$). This is detailed at the end of Sect. 3.3 (line 420).

    The forward and the reverse interferograms only systematically differ because of electrical dispersion (common-mode) and optical dispersion (differential-mode, see phases shown in Fig. 13b, this includes the effect of a shifted sampling grid). These phases are corrected through division by the complex spectral gain estimate during radiometric calibration (Eq. 11). The FFOV correction and the resampling compensate for instrument line shape effects instead associated to the radiant intensity inside the interferometer and the effective modulation frequency of the metrology laser. There is no conceptual or empirical evidence to suggest that those properties could depend on the scan direction. In any case the characterization of the associated parameters is always performed on reverse-forward averaged data, so what is extracted

is a global parameter that would mask any potential variation, correcting the distortion to first order and leaving potential second-order distortions uncorrected.

Although we are only summarizing the content of Section 3 at this point of the text and prefer not to dwell into the details, we agree that the sentence pointed out by the reviewer can be clarified and that hints about the reasoning can be given:

(page 16, line 269) The forward and reverse scan directions are treated separately until completion of the radiometric calibration, which allows for the electrical and optical dispersion to be treated properly. The resulting radiometrically-calibrated spectra from each scan direction are then averaged together. Finally, the averaged spectrum is further processed to correct for distortions that are independent of scan direction (finite field of view, non-standard spectral grid) and it is cropped to remove the samples that are outside the ASSIST's sensitivity range.

We also modified Fig. 15 and its caption to show the spectra associated to both reverse and forward scan directions so that the reader can better see where they are identical and where they differ (because of random measurement noise):

[Figure]

"**Figure 15.** Typical calibrated complex  spectra measured with ASSIST-22  for the two scan directions. (a) Real part. The forward scan direction spectrum is traced on top of the reverse scan direction spectrum. Planck curves at different temperatures are also illustrated: 60°C (HBB), -100°C, and 281.26 K (representative of the temperature of the air in the first few meters above the instrument, the simultaneous mast sensor readout was $T_{ext} = 281.3$ K). (b) Imaginary part. RU = mW (m$^2$ sr cm$^{-1}$)$^{-1}$."

27. *(page 20, line 335) Do you perhaps need to move the commas? For example: Fitted values of the factor $2a_2V_{0,Hd}$, for HBB views normally associated to the highest photon irradiance for a dozen ASSIST instruments, are distributed around 0.10 with extreme values reaching 0.04 and 0.16.*
   **Response:** We agree the sentence is confusing, but instead suggest the following reorganization:

(page 20, line 335) Fitted values of the factor $2a_2V_{0,Hd}$ (H subscript for the HBB views), which is  associated to the highest photon irradiance,  are uniformly distributed  as 0.10±0.06 for a dozen ASSIST units, .

28. *(page 20, line 338) Remove the ( ) around the reverse scan description and write it as a stand-alone sentence.*
**Response:** Agreed. The correction has been made.

29. *(page 25-26, Fig. 16) As no two days are the same, and no two sites the same, is there any particular relevance to showing these signals evolving over a day? Showing cases near the limits might provide more information.*
**Response:** The reviewer is right in pointing out that no two days are the same and that the data shown in Fig. 16 are not particularly special. However, our objectives with this figure were a) to show that the responsivities remain stable despite the relatively large excursions of the ABB temperature ($T_A$) and the reflected temperature ($T_r$) throughout the day, a hallmark of an accurate radiometric calibration, b) to show that HBB temperature ($T_H$) and reference temperature ($T_R$) do not fluctuate, and c) to show what happens when precipitation is detected by the instrument. The ASSIST-22 dataset from 2024-06-16 was convenient to attain those three objectives. We recognize this should be stated explicitly to justify presentation of this figure:

(page 25, line 423) Figure 16 illustrates how the radiometric calibration parameters typically evolve over a full day of operation. Despite a 14°C variation in external air temperature and a 19°C variation in ABB temperature during that day, the responsivity extracted through radiometric calibration remains stable, implying accurate modeling of effective temperatures and emissivities. Moreover, the HBB and reference port temperatures do not fluctuate significantly around their setpoint values.

30. *(page 26, line 427) Something doesn't make sense in the wording.*
**Response:** Agreed. At least one word is missing in the sentence, but the whole paragraph is difficult to read. We modified as follows:

(page 26, line 426) A monochromatic ($\nu_c$) radiation source filling the instrument's finite FOV does not yield a pure cosine interferogram oscillating at $\nu_c$  (Genest and Tremblay, 1999).  Instead, since the modulation frequency is proportional to the cosine of  a ray's propagation angle  inside the interferometer, the spectrum produced by the instrument is distorted: it  shows a response distributed between $\nu_c \cos\theta_{max}$ and $\nu_c \cos\theta_{min}$ instead of a single infinitesimal component at $\nu_c$, where $\theta_{max}$ and $\theta_{min}$ are the maximum and minimum allowed propagation angles  relative to the optical axis . The spectral resolution is thus limited further than  predicted from the interferogram's implicit truncation alone (finite maximum OPD or boxcar apodization) .

31. *(page 28, Fig. 17) I am unsure how to interpret the value of the "first moment". Is that the median of the area under the curve? Center of mass means what in this context?*
**Response:** We agree this can be clarified and therefore replaced "first moment" with "mean" and eliminated allusions to the concept of "center of mass" (a facultative mechanical analogy):

(page 27, line 454) Furthermore, since the boxcar's  mean  is located at…

(page 28, Fig. 17) The dotted vertical lines are the associated  distribution means and the dot-dashed purple curve is the effective boxcar FFOV ILS at 1250 cm$^{-1}$ (same  mean), with $\bar{b}$ = 23 mrad like in the top panel.

32. *(page 29, line 464) "… the relative spectral shift is spectrally dependent" … does this mean that the spectral shift varies with wavelength? Some other characteristic of the spectrum?*
**Response:** The spectral shift varies with wavelength or wavenumber. We recognize the phrasing is ambiguous and we have modified the text as follows:

(page 29, line 463) The  mean of the FFOV ILS varies with normalized wavenumber (Fig. 17, middle); the corollary is that the relative spectral shift  varies with wavenumber, causing a non-uniform compression of the  spectral axis.

33. *(page 29, line 472) Perhaps you could comment briefly on the most critical facet of the detector alignment as determined by your new procedure?*
**Response:** Agreed:

(page 29, line 471) There used to be a large spread in the values of $\bar{b}$, from 20 to 30 mrad near 1250 cm$^{-1}$ for different ASSIST instruments , but the detector alignment procedure  was refined in 2023 to tighten the distribution  to the 21 to 24 mrad range, primarily by performing rapid characterizations on the production line to enhance feedback during the fine-tuning of the detector focus.

34. *(page 30, Fig. 18) Is the radiance represented here actual measurements from the ASSIST-22, or modeled radiance based on the characteristics of ASSIST-22?*
**Response:** Fig. 18 shows actual measurements from ASSIST-22. Except for figures 10, 12, and 17 which were created from models, the data shown in this manuscript originate from actual ASSIST measurements. We agree this should be made explicit and have modified the captions accordingly:

(page 15, Fig. 9) Figure 9. Typical calibration module temperatures measured over a full day of uninterrupted outdoor operation…

(page 19, Fig. 10) Figure 10. Modeled spectrum of the elementary interferogram…

(page 21, Fig. 11) Figure 11. Typical impact of the NLC in the HgCdTe channel measured with ASSIST-22…  Here, $a_2$ = …

(page 23, Fig. 13) Figure 13. Typical complex spectral gain measured with ASSIST-22…

(page 24, Fig. 14) Figure 14. Typical complex spectral offset measured with ASSIST-22…

(page 25, Fig. 15) Figure 15. Typical calibrated complex  spectra measured with ASSIST-22  for the two scan directions…

(page 26, Fig. 16) Figure 16. (a) Typical temperatures involved in radiometric calibration measured over a full day of uninterrupted outdoor operation…

(page 28, Fig. 17) Figure 17. FFOV instrument line shape model for the HgCdTe channel….

(page 30, Fig. 18) Figure 18. Typical impact of the FFOV broadening correction measured with ASSIST-22 ( with $\bar{b}$ = 27.0 mrad in the HgCdTe channel after a +182 ppm stretching operation)…

(page 31, Fig. 19) Figure 19. Typical impact of the resampling correction measured with ASSIST-22 ( with $v'_s$ = 15799.60 cm$^{-1}$ in the HgCdTe channel, for a -38 ppm resampling operation)…

(page 32, Fig. 20) Figure 20. (a) Typical final radiance in the HgCdTe channel and for a clear sky view  measured with ASSIST-22 (medium nonlinearity, high FFOV-distortion)…

(page 33, Fig.21) Figure 21. (a) Typical final radiance in the InSb channel and for a clear sky view  measured with ASSIST-22 (high FFOV-distortion)…

(page 34, Fig. 22) Figure 22. Typical "sky NEN" measured with ASSIST-22 and calculated as…

35. *(page 30, line 492) "typically displays strong out-of-band noise" how is this happening? Are there not optical filters, electronic filters? Or what is being referred to here as out-of-band noise?*
    **Response:** This segment refers to the state of the radiance vector after radiometric calibration. For those spectral samples where the spectroradiometer is unresponsive ("out-of-band"), the complex spectrum is essentially pure complex noise for all scenes (see Fig. 11 for an example). From Eq. 7a, the out-of-band spectral gain is pure complex noise as well. When performing the radiometric calibration as per Eq. 11, the complex spectrum associated to the sky scene is divided by the spectral gain and its out-of-band noise is therefore amplified. This effect cannot be controlled through optical and/or electronic filtering, and it is not detrimental or problematic unless the spectral samples are scrambled. But FFOV correction and spectral resampling do scramble the spectral samples. We agree this can be clarified and have made the following modification:

    (page 30, line 491) Outside the sensitivity range ("out-of-band") and under strong internal absorption lines, the radiance estimate $\hat{L}_S$  displays strong  noise because of the division by a low or purely random spectral gain during radiometric calibration (Eq. 11) (Rowe et al., 2011a). It is therefore good practice…

36. *(page 34, Fig. 22) Looks like a blue and red curve on my screen. You might want to refer to line type or some other line or plot characteristic to avoid color confusion.*
    **Response:** (see item 13 above) The up-to-date version of Fig. 22 and its caption (shown below) should address the reviewer's valid concern.

[Figure]

"**Figure 22**. Typical "sky NEN" measured with ASSIST-22 and calculated as a moving standard deviation over 52 spectral samples (25 cm$^{-1}$-wide bins) of the imaginary radiance $\widehat{D}_S''$. The illustrated dotted curves are similar to the NESR calculated for a 12.6 s dwell time and 0.48 cm$^{-1}$ bin spacing and depend on fluctuating sky and ABB radiances. The smooth curve represents 1 % of a 293 K Planck curve. RU = mW (m$^2$ sr cm$^{-1}$)$^{-1}$.""